# East Asian-specific and cross-ancestry genome-wide meta-analyses provide mechanistic insights into peptic ulcer disease

Yunye He [1], Masaru Koido [1], Yoichi Sutoh[2], Mingyang Shi[1], Yayoi Otsuka-Yamasaki[2], Hans Markus Munter [3], BioBank Japan*, Takayuki Morisaki[4,5], Akiko Nagai[6], Yoshinori Murakami [4], Chizu Tanikawa[5], Tsuyoshi Hachiya [2], Koichi Matsuda [5], Atsushi Shimizu[2] & Yoichiro Kamatani [1]✉

Peptic ulcer disease (PUD) refers to acid-induced injury of the digestive tract, occurring mainly in the stomach (gastric ulcer (GU)) or duodenum (duodenal ulcer (DU)). In the present study, we conducted a large-scale, cross-ancestry meta-analysis of PUD combining genome-wide association studies with Japanese and European studies (52,032 cases and 905,344 controls), and discovered 25 new loci highly concordant across ancestries. An examination of GU and DU genetic architecture demonstrated that GUs shared the same risk loci as DUs, although with smaller genetic effect sizes and higher polygenicity than DUs, indicating higher heterogeneity of GUs. *Helicobacter pylori* (HP)-stratified analysis found an HP-related host genetic locus. Integrative analyses using bulk and single-cell transcriptome profiles highlighted the genetic factors of PUD being enriched in the highly expressed genes in stomach tissues, especially in somatostatin-producing D cells. Our results provide genetic evidence that gastrointestinal cell differentiations and hormone regulations are critical in PUD etiology.

PUD refers to the acid-induced injury of the digestive tract, occurring mainly in the stomach (GU) or proximal segment of the duodenum (DU) with bleeding, perforation or gastric outlet obstruction as the major complication. PUD is one of the most common gastrointestinal disorders, with a lifetime prevalence rate of approximately 5–10% in the general population[1]. The prevalence of PUD has been reported to be substantially higher in east Asians (EAS) than Europeans (EUR)[1], with GU being more common than DU in the Japanese population and DU being more common in Europeans[2].

With HP infection and the use of nonsteroidal anti-inflammatory drugs (NSAIDs) being two of the most common causes of GUs and

DUs[3], genetic factors also play a critical role in the development of PUD[4]. Previous genome-wide association studies (GWASs) of PUD had identified multiple loci, mainly HP related, in Europeans[5,6]. Given the relatively high prevalence of PUD and HP infection in east Asians and the remarkably limited number of risk loci identified in east Asian populations[2,7], GWASs with a larger sample size of EAS ancestry individuals would be required to enhance our understanding of genetic etiology of PUD. As GUs and DUs differ in various aspects, such as the proportion of ulcers that are attributable to HP infection[8], the genetic differences across PUD subtypes and the key cell types involved in their etiology should be investigated. Epidemiological studies have

[1]Laboratory of Complex Trait Genomics, Graduate School of Frontier Sciences, The University of Tokyo, Tokyo, Japan. [2]Iwate Tohoku Medical Megabank Organization, Iwate Medical University, Iwate, Japan. [3]Victor Phillip Dahdaleh Institute of Genomic Medicine and Department of Human Genetics, McGill University, Montreal, Québec, Canada. [4]Division of Molecular Pathology, Institute of Medical Science, The University of Tokyo, Tokyo, Japan. [5]Laboratory of Clinical Genome Sequencing, Graduate School of Frontier Sciences, The University of Tokyo, Tokyo, Japan. [6]Department of Public Policy, Institute of Medical Sciences, The University of Tokyo, Tokyo, Japan. *A list of authors and their affiliations appears at the end of the paper. ✉e-mail: kamatani.yoichiro@edu.k.u-tokyo.ac.jp

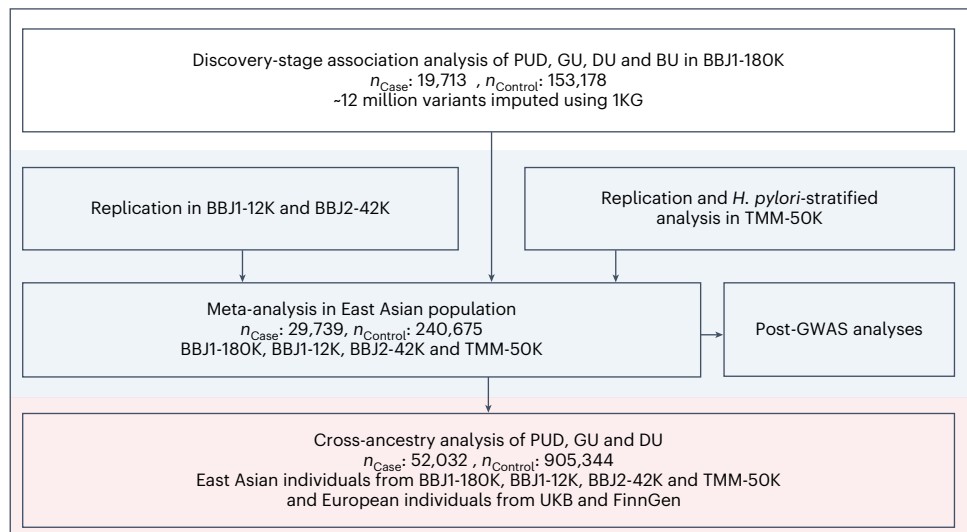

**Fig. 1 | Overview of the three-stage study design.** The 1000 Genomes Project reference panel (1KG Phase 3). BBJ1-180K consists of approximately 180,000 individuals from BBJ1. BBJ1-12K consists of approximately 12,000 individuals from BBJ1. BBJ2-42K consists of approximately 42,000 individuals from BBJ2. TMM-50K consists of approximately 50,000 individuals from Tohoku Medical Megabank.

suggested that DU is a protective factor against gastric cancer (GC)[9]. However, whether genetic factors for PUD and GC are concordant and can explain the epidemiological findings remain unclear. Therefore, we anticipated that large-scale genetic studies of PUD could not only expand our understanding of PUD biology but also provide insights into the genetic factors that interact with HP or lead to different outcomes of PUD or GC, potentially enabling a more accurate prediction of individual risk in clinical settings.

To address these issues, we conducted a large east Asian-specific and cross-ancestry genome-wide analysis of PUD and PUD subtypes along with four Japanese studies and two European cohorts totaling 52,032 PUD cases and 905,344 controls.

## Results

We conducted a three-stage genome-wide analysis of PUD and its subtypes. An overview of the workflow is provided in Fig. 1 and Supplementary Fig. 1. PUD cases in the east Asian populations were obtained by combining individuals with any of the two major PUD subtypes (DU and GU), which were classified based on the anatomical sites where peptic ulcers occurred. Individuals with comorbidities of GU and DU were classified as BU (both GU and DU) cases (Methods, Supplementary Fig. 2 and Supplementary Table 1).

### Association analyses of PUD and its subtypes

First, we performed GWASs of PUD and PUD subtypes (DU, GU and BU) in the discovery stage on the BioBank Japan first cohort (BBJ1)[10]-180,000 (180K) dataset (Methods). The dataset included 19,713 cases of PUD and 153,178 controls of east Asian ancestry and was imputed using the 1000 Genomes Project Phase 3 (ref. 11) (1KG Phase 3) reference panel. A total of 13,846,852 variants (minor allele count (MAC) >20 and $R^2 > 0.3$) were tested in east Asians for association with a generalized linear mixed model using SAIGE[12], which controls for the case–control imbalance (case-to-control ratios ranged from 1:7.7 to 1:82 in BBJ1-180K; Supplementary Table 1). For PUD, ten genome-wide significant loci ($P < 5.0 \times 10^{-8}$) were identified, five of which had not been reported as genome-wide significant loci in previous GWASs of PUD or any subtype. In addition, 14 loci reached the significance threshold for DU, including 7 new loci (3 of which overlapped with new PUD loci). One previously reported locus at *PSCA*[2] was identified for GU and BU. A total of 15 nonoverlapping genetic loci (Methods) reached the significance threshold for PUD or any subtype, of which 9 were new

(Supplementary Table 2 and Supplementary Fig. 3). Analysis of the X chromosome identified one known locus at *GUCY2F*[7] for PUD, DU and GU (Supplementary Table 3). Thirteen nonoverlapping significant loci were identified in sex-stratified analysis (thirteen for men and one for women; Supplementary Table 3).

Replication was conducted in individuals from three independent studies, namely BBJ1-12K (1,001 cases), BBJ2-42K (3,637 cases) and TMM[13]-50K (a population-based study (5,388 cases); Supplementary Table 1 and Methods). The replication datasets were imputed using the 1KG Phase 3 panel and tested for associations of autosomal variants with the same settings as in discovery GWASs. Among the nine new lead variants associated with PUD or subtypes, four were nominally associated ($P < 0.05$) with PUD or its subtypes in the same direction in at least two replication datasets. Notably, five new loci were replicated in the population-based dataset ($P < 0.05$ in the same direction; Supplementary Table 4).

Next, we performed an east Asian-specific meta-analysis combining the discovery GWASs and three replication GWASs ($n_{case}$ = 29,739; $n_{control}$ = 240,675). Fixed-effect meta-analyses using the inverse-variance weighted (IVW) method were performed for PUD and PUD subtypes. The genomic inflation factors ($\lambda_{gc}$) and linkage disequilibrium (LD) score regression (LDSC)[14] intercepts ranged from 1.03 to 1.08 and from 1.01 to 1.02, respectively (Supplementary Table 5), indicating no substantial bias. In the EAS-specific meta-analysis, we detected 25 nonoverlapping risk loci associated with PUD or any subtype, including 11 additional new loci (Table 1, Supplementary Fig. 4 and Supplementary Table 6).

Finally, we collected publicly available European GWASs of PUD and its subtypes using samples from FinnGen and UK Biobank (UKB)[5,12,15] (Supplementary Table 7). After quality control (QC) and harmonization (Methods), a fixed-effect, IVW, cross-ancestry meta-analysis (52,032 cases of PUD and 905,344 controls) was performed, combining the Japanese and European studies. Six additional loci for PUD and DU reached the genome-wide significance level ($P < 5.0 \times 10^{-8}$; Table 1, Fig. 2, Supplementary Table 8 and Supplementary Fig. 5). Furthermore, we performed a cross-ancestry meta-regression utilizing MR-MEGA (Meta-Regression of Multi-AncEstry Genetic Association)[16] and identified 23 known and described new loci mentioned above in the east Asian-specific and cross-ancestry meta-analyses (Supplementary Table 9). In total, we identified 25 nonoverlapping new loci for PUD and its subtypes in the east Asian-specific and cross-ancestry meta-analyses

**Table 1 | Significant loci associated with PUD or PUD subtypes from genome-wide meta-analyses**

| Phenotypes | rs ID | Chromosome | Position (GRCh37) | Nearest gene | Effect allele/ noneffect allele | OR | 95% CI | P | Analysis | EAF$_{EAS}$ | EAF$_{EUR}$ |
|---|---|---|---|---|---|---|---|---|---|---|---|
| PUD | rs59781317 | 1 | 156016356 | UBQLN4 | G/A | 1.04 | (1.03–1.06) | $3.51 \times 10^{-8}$ | Cross-ancestry | 0.77 | 0.25 |
| PUD | rs184426772 | 2 | 182502860 | CERKL | C/T | 2.22 | (1.68–2.92) | $1.61 \times 10^{-8}$ | EAS | 0.01 | 0.00 |
| PUD, DU | rs11692085 | 2 | 219963550 | NHEJ1 | T/C | 1.05 | (1.04–1.07) | $2.58 \times 10^{-12}$ | Cross-ancestry | 0.27 | 0.66 |
| PUD, DU, GU | rs4685405 | 3 | 16981683 | PLCL2 | T/G | 1.05 | (1.04–1.07) | $9.84 \times 10^{-12}$ | Cross-ancestry | 0.52 | 0.18 |
| DU, PUD, GU | rs79928271 | 3 | 169139475 | MECOM | A/T | 0.90 | (0.88–0.93) | $2.02 \times 10^{-11}$ | EAS | 0.28 | 0.13 |
| DU | rs13086914 | 3 | 171491650 | PLD1 | G/T | 0.91 | (0.88–0.94) | $4.83 \times 10^{-8}$ | EAS | 0.20 | 0.39 |
| DU | rs34742353 | 4 | 47634802 | CORIN | A/AT | 1.11 | (1.07–1.15) | $5.88 \times 10^{-9}$ | EAS | 0.83 | 0.74 |
| PUD | rs2553380 | 4 | 124538120 | LINC01091 | C/T | 0.94 | (0.92–0.96) | $1.00 \times 10^{-8}$ | EAS | 0.65 | 0.84 |
| PUD, DU, GU | rs3805497 | 5 | 40746885 | TTC33 | T/A | 1.08 | (1.06–1.09) | $5.32 \times 10^{-25}$ | Cross-ancestry | 0.58 | 0.25 |
| PUD | rs1801020 | 5 | 176836532 | F12 | G/A | 0.95 | (0.94–0.97) | $1.38 \times 10^{-11}$ | Cross-ancestry | 0.35 | 0.75 |
| DU | rs41265804 | 6 | 29924159 | HLA-A | G/C | 0.92 | (0.90–0.95) | $1.00 \times 10^{-9}$ | Cross-ancestry | 0.49 | 0.17 |
| PUD | rs146095444 | 6 | 131582218 | AKAP7 | T/C | 1.24 | (1.15–1.34) | $2.76 \times 10^{-8}$ | EAS | 0.02 | 0.00 |
| PUD, DU | rs416879 | 6 | 160774838 | SLC22A3 | G/A | 0.96 | (0.95–0.97) | $2.48 \times 10^{-9}$ | Cross-ancestry | 0.28 | 0.48 |
| PUD, DU, GU, BU | rs72607744 | 7 | 127560541 | SND1 | A/G | 1.16 | (1.12–1.20) | $4.45 \times 10^{-15}$ | EAS | 0.10 | 0.00 |
| DU | rs7470279 | 9 | 80607789 | GNAQ | T/A | 1.09 | (1.06–1.12) | $1.56 \times 10^{-10}$ | Cross-ancestry | 0.37 | 0.15 |
| PUD | rs10992997 | 9 | 96701663 | BARX1 | G/A | 1.05 | (1.03–1.06) | $1.41 \times 10^{-8}$ | Cross-ancestry | 0.14 | 0.42 |
| DU | rs3019776 | 11 | 68826155 | TPCN2 | G/A | 1.10 | (1.07–1.13) | $1.38 \times 10^{-9}$ | Cross-ancestry | 0.53 | 0.97 |
| BU | rs147272036 | 14 | 95298230 | GSC-DT | G/C | 3.77 | (2.36–6.02) | $2.85 \times 10^{-8}$ | EAS | 0.01 | 0.00 |
| DU | rs511893 | 16 | 88990941 | CBFA2T3 | G/T | 0.91 | (0.88–0.94) | $6.39 \times 10^{-9}$ | Cross-ancestry | 0.30 | 0.34 |
| DU | rs2642030 | 17 | 65605075 | PITPNC1 | G/A | 1.07 | (1.05–1.10) | $2.00 \times 10^{-8}$ | Cross-ancestry | 0.56 | 0.25 |
| DU | rs6117384 | 20 | 6673542 | LINC01713 | C/T | 1.09 | (1.07–1.12) | $6.32 \times 10^{-12}$ | Cross-ancestry | 0.74 | 0.73 |
| PUD, DU, GU, BU | rs6123837 | 20 | 57465571 | GNAS | A/G | 1.05 | (1.04–1.07) | $2.12 \times 10^{-14}$ | Cross-ancestry | 0.58 | 0.36 |
| PUD, DU | rs12625329 | 20 | 62709274 | RGS19 | A/G | 1.04 | (1.03–1.06) | $8.09 \times 10^{-9}$ | Cross-ancestry | 0.66 | 0.46 |
| PUD, DU, GU | rs11416248 | 22 | 25008477 | GGT1 | CT/C | 0.93 | (0.91–0.95) | $2.28 \times 10^{-12}$ | EAS | 0.32 | 0.22 |
| DU | rs7288137 | 22 | 50458020 | TTLL8 | A/G | 1.08 | (1.06–1.12) | $7.73 \times 10^{-9}$ | Cross-ancestry | 0.20 | 0.22 |
| **Previously known loci** | | | | | | | | | | | |
| PUD, DU | rs1345894981 (rs147048677) | 1 | 155161794 | MUC1 | T/C | 1.14 | (1.10–1.19) | $2.54 \times 10^{-12}$ | Cross-ancestry | 0.02 | 0.06 |
| DU, PUD, GU, BU | rs2294008 | 8 | 143761931 | PSCA | T/C | 0.66 | (0.64–0.68) | $7.69 \times 10^{-189}$ | EAS | 0.64 | 0.46 |
| PUD, DU, GU | rs8176719 | 9 | 136132908 | ABO | TC/T | 0.93 | (0.92–0.95) | $5.76 \times 10^{-24}$ | Cross-ancestry | 0.45 | 0.37 |
| PUD | rs78459074 | 11 | 1029905 | MUC6 | G/A | 0.95 | (0.93–0.96) | $7.60 \times 10^{-9}$ | Cross-ancestry | 0.21 | 0.12 |
| PUD, DU | rs10500661 | 11 | 6273744 | CCKBR | C/T | 1.11 | (1.09–1.13) | $7.75 \times 10^{-27}$ | Cross-ancestry | 0.09 | 0.20 |
| PUD, DU,GU | rs9581957 | 13 | 28557889 | URAD | T/C | 1.07 | (1.06–1.09) | $2.20 \times 10^{-20}$ | Cross-ancestry | 0.18 | 0.33 |
| PUD, DU,GU | rs34074411 | 17 | 39867248 | GAST | T/C | 1.07 | (1.06–1.09) | $1.07 \times 10^{-22}$ | Cross-ancestry | 0.63 | 0.45 |
| DU, PUD | rs11665674 | 19 | 49196275 | FUT2 | G/A | 1.15 | (1.11–1.18) | $3.38 \times 10^{-17}$ | EAS | 0.45 | 0.00 |

Lead variants from each significant locus identified in the fixed-effect population-specific or the cross-ancestry meta-analysis are shown. Only the most significant lead variant is shown for loci associated with multiple phenotypes or identified in multiple analyses. Variants were annotated with the nearest genes. EAF$_{EAS}$ and EAF$_{EUR}$, effect allele frequencies in population-specific meta-analysis. For variants not available in the datasets, EAF was obtained from the Genome Aggregation Database or the 1000 Genomes Project.

(Supplementary Fig. 6), although one new locus identified in the discovery stage was not significant in any of the meta-analyses (Supplementary Table 2).

**Cross-ancestry comparison**

With the large available datasets for PUD and its subtypes in EAS, we investigated the shared and distinct risk loci for PUD in EAS and EUR individuals. We compared the per-allele effect sizes of lead variants associated with PUD or any of the subtypes available for both ancestries (Methods). The effect sizes for PUD showed a relatively high correlation (27 variants with minor allele frequency (MAF) > 0.01 in both populations; $r = 0.79$) between the two ancestries, although we detected 9 variants (9 out of 27 effect-size differences (difference in log(odds ratio (OR)) > 0.05; Fig. 3a and Supplementary Table 10). The high correlation remained after the winner's curse (WC) corrections (Methods, Supplementary Table 11 and Supplementary Fig. 7). To further examine the difference in genetic architecture of PUD between east Asians and Europeans, we conducted a cross-ancestry genetic correlation analysis

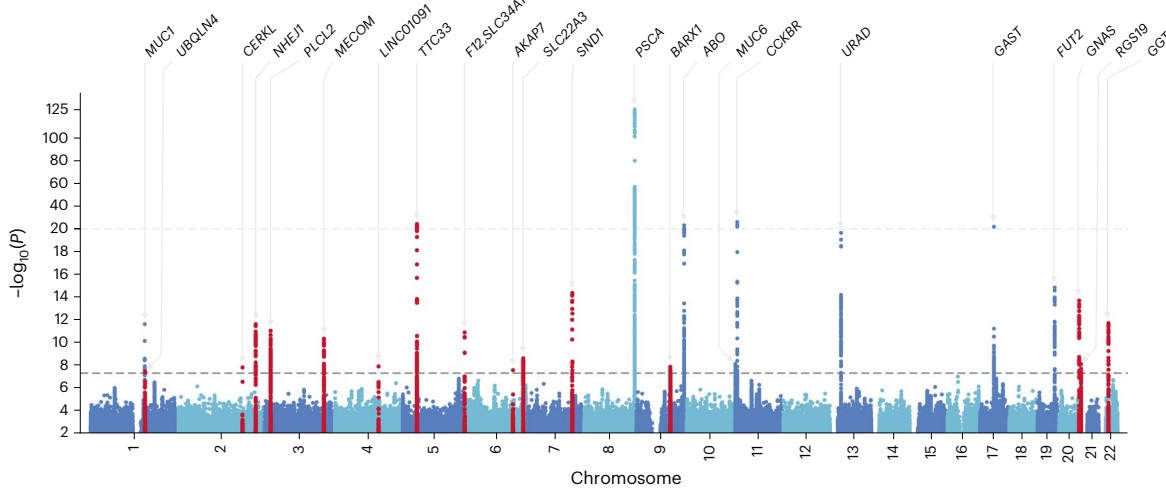

**Fig. 2 | Manhattan plot of the cross-ancestry meta-analysis for PUD.** Two-sided *P* values were derived from the cross-ancestry meta-analysis of 52,032 cases and 905,344 controls of EAS or EUR ancestry. Meta-analysis was performed using the IVW method under the fixed-effect model. For variants above the top light-gray dashed line ($-\log_{10}(P) > 20$), values are rescaled. Lead variants are annotated with the nearest gene name(s). New loci are highlighted in red. Variants are plotted against GRCh37 (hg19). The bottom dark-gray dashed line indicates the genome-wide significance threshold ($P < 5.0 \times 10^{-8}$). Variants with $-\log_{10}(P) < 2$ were omitted.

using Popcorn[17] (Methods). The genetic impact was significantly different from 1 (null hypothesis: $\rho_{gi} = 1$) for PUD (Fig. 3b; genetic impact correlation $\rho_{gi} = 0.65$, $P = 3.0 \times 10^{-4}$), indicating the difference in genetic architecture of PUD across ancestries. For the subtypes, effect sizes for DU showed a higher correlation ($r = 0.79$) across ancestries compared with that for GU ($r = 0.63$; Supplementary Fig. 8 and Supplementary Table 10). The genetic correlation of GU was relatively low ($\rho_{gi} = 0.45$, $P = 7.3 \times 10^{-3}$), whereas the genetic architecture of DU did not show a significant difference across ancestries ($\rho_{gi} = 0.72$, $P = 9.6 \times 10^{-2}$; Supplementary Table 12).

### Characterization of PUD-associated loci in east Asians

To explore the secondary signals at the identified loci, we conducted a stepwise conditional analysis using COJO[18] with an in-sample LD reference for EAS (Methods). We detected four additional independent signals reaching genome-wide significance ($P < 5.0 \times 10^{-8}$) for PUD and three independent signals at the *PSCA* locus for DU (Supplementary Table 13). The *PSCA* locus had the largest number of independent associations (three for PUD, four for DU and two for GU and BU). Near the *CDX2* and *GAST* loci, two of the previously reported loci in European individuals[5], we detected independent signals at *PDX1* (Fig. 4a) and *JUP2* (Fig. 4b) loci, respectively (Supplementary Tables 13 and 14).

We conducted a fine-mapping analysis using SuSiE[19] to identify the causal variants. We searched for nonsynonymous variants in 95% credible sets to link the disease-associated loci to potential alteration of protein functions. A total of ten nonsynonymous variants at six nonoverlapping loci were identified, six of which were in new loci for PUD and its subtypes (Supplementary Table 15). Of those, rs2233580 (PAX4; p.R200H; combined annotation-dependent depletion (CADD)[20] score = 29.8) was also associated with type 2 diabetes. The variant was common (MAF > 0.05) in 1KG EAS but almost monozygotic in non-EAS populations. Rs4745 (EFNA1; p.D159V; posterior inclusion probability (PIP) = 0.05 for DU; CADD score = 15.2) was common in EAS and EUR and associated with GC. This was the lead variant of *cis*-splicing quantitative trait loci (sQTLs) for *EFNA1* in the stomach and was in high LD with rs4072037 (ref. 21) (lead sQTL variant for *MUC1* in the stomach; LD $r^2 = 0.74$ in 1KG EAS). In addition to the missense variants, we found rs4390169 in the credible set (upstream of *EFNA1*; PIP = 0.06 for DU; in high LD with rs4745, LD $r^2 = 0.99$ in 1KG EAS and EUR) to be the lead variant of *cis*-protein QTL (pQTL) in plasma for *EFNA1* (ref. 22).

In the credible sets of previously reported *ABO* and *FUT2* loci for PUD[2,5], we identified rs8176719 (lead variant at *ABO* locus) and rs1047781 (in the credible sets at *FUT2* locus; PIP = 0.63). Deletion of rs8176719 resulted in the O-allele, whereas rs1047781 (p.I140F) was an EAS-specific common variant (MAF = 0.439 in 1KG EAS) and its A-allele determined the FUT2 secretor status. We performed a logistic regression analysis to investigate the correlation of ABO blood group and FUT2 secretor status with PUD. Blood group O (OR = 1.14, $P = 6.0 \times 10^{-14}$) and nonsecretor status (OR = 1.17, $P = 2.9 \times 10^{-11}$) were significantly correlated with a higher risk of PUD, which was consistent for all PUD subtypes (Supplementary Table 16). To investigate the potential interactions between blood group O and nonsecretor status, logistic regression analysis including an interaction term was performed. However, significant interactions ($P < 0.05/8$, Bonferroni's correction) were not detected between blood group O and nonsecretor status (Supplementary Table 17).

### Overlap of eQTL and pQTL with risk variants for PUD

To detect the functionally relevant genes, we searched the Genotype-Tissue Expression (GTEx) v.8 datasets[21] for overlap of lead *cis*-expression QTLs (eQTLs) with PUD signals or their LD proxies (LD $r^2 > 0.6$ in 1KG EAS or EUR)[23]. The most significant eQTL hits for a gene within each tissue type were interpreted (Supplementary Fig. 9). We identified an overlap of new variants with eQTLs associated with *IHH*, *PLCL2*, *PTGER4*, *ZNF322*, *HIATL1*, *FAM211B* and *GGT1* in the stomach.

We searched five recent large-scale pQTL datasets[22,24–27] from serum or plasma for overlap of *cis*- or *trans*-pQTL with PUD signals or their LD proxies. We observed overlaps with 88 unique significant pQTL associations, most of which (93.1%) were *trans*-pQTL and involved the lead SNP at the *ABO* locus (Supplementary Table 18). The *cis*-pQTL alleles in LD with PUD risk alleles were associated with increased levels of EFNA1 and OBP2B, and decreased levels of NHEJ1, ABO and GGT1 (the *cis*-pQTLs overlapped with the *cis*-eQTLs mentioned above for *EFNA1, OBP2B, NHEJ1* and *GGT1*). For *trans*-pQTLs in LD with the lead variants, we observed links with multiple proteins, including F8, F10, PROS1 (blood coagulation related) and trefoil factor family peptides (which play important roles in response to gastrointestinal mucosal injury). Additional analysis suggested plausible proteins and pathways (Supplementary Note and Supplementary Table 19).

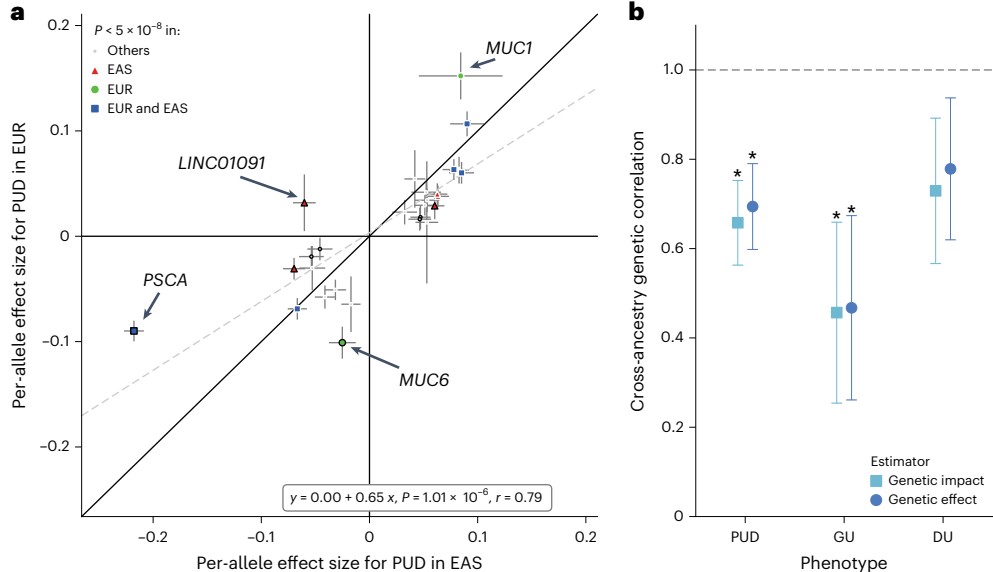

**Fig. 3 | Cross-ancestry effect-size comparison and genetic correlation analysis. a**, Per-allele effect-size (logarithm of ORs) comparison using east Asian-specific and European-specific summary statistics for PUD. Lead variants associated with PUD or any subtype in east Asian-specific, European-specific or cross-ancestry meta-analysis were selected for comparison. Two-sided *P* values were derived from the ancestry-specific meta-analyses. The most significant associations were shown if overlapping variants existed (interval <500 kb). Data are presented as effect-size estimates (log(OR)) ± s.e.m. Variants with nominally significant heterogeneity (Cochran's *Q* test; two-sided $P_{het} < 0.05$) were denoted by the black marker edges. The gray dashed line represents the fitted linear regression line with annotation at the bottom right (*P* value is derived from a two-sided Student's *t*-test for the slope). Pearson's *r* is shown. **b**, Cross-ancestry genetic correlation for PUD, GU and DU estimated by Popcorn. The gray dashed line indicated the null hypothesis (ρ = 1). Asterisks indicate estimates that were significantly <1 (two sided; FDR < 5%). Data are presented as genetic correlation estimates ± s.e.m. Sample sizes are provided in Supplementary Table 12.

## Genetic correlation and pleiotropic effects

We conducted cross-trait LD score regression[28] to evaluate the genetic correlation across PUD-related traits (Fig. 5a and Supplementary Table 20). DU and GU showed significantly high genetic correlations ($r_g = 0.79$, false discovery rate (FDR) <5%) with each other, as expected. Although not statistically significant (FDR < 5%), GU showed a positive genetic correlation with GC ($r_g = 0.17$), whereas DU was negatively correlated ($r_g = -0.14$). We also investigated the genetic correlation of PUD with dietary habits[29] and complex traits in BBJ[30,31] (Methods); no significant genetic correlation was observed between PUD and other complex traits in EAS (FDR < 5%; Supplementary Figs. 10–12 and Supplementary Table 21).

To investigate the pleiotropic effects of distinct variants, we performed a phenome-wide association study (PheWAS) lookup using previous large-scale GWASs in a Japanese population[7]. Among the 27 available lead variants associated with PUD and its subtypes, 16 reached the genome-wide significance threshold for at least one trait ($P < 5.0 \times 10^{-8}$). From them, 12 variants were associated with at least 2 traits after Bonferroni's correction ($P < 8.6 \times 10^{-6}$; Supplementary Figs. 13–15 and Supplementary Table 22). Both type 2 diabetes (two at *SND1–PAX4* locus and one at *GAST* locus) and GC (*EFNA1*, *PTGER4* and *PSCA* loci) shared three significant variants after Bonferroni's correction ($P < 8.6 \times 10^{-6}$) with PUD or its subtypes (Supplementary Fig. 16).

## HP-stratified analysis

To examine the differences in genetic architectures between HP-induced and HP-unrelated peptic ulcers, we conducted HP-stratified association tests for PUD in HP⁺ and HP⁻ individuals from TMM-50K (Methods and Supplementary Table 23). For the distinct PUD signals identified in the EAS population (29 variants with MAF > 0.01), per-allele effect sizes for PUD between HP⁻ and HP⁺ status were highly correlated (Fig. 5b; slope = 0.84, s.e.m. = 0.07, r = 0.91; Supplementary Fig. 17 and Supplementary Table 24). We identified one lead SNP (rs12792379), specifically associated with HP⁺ PUD, at *CCKBR* (OR = 1.18, 95% confidence

interval (CI) = 1.05–1.34 for HP⁺ PUD; OR = 1.01, 95% CI = 0.92–0.11 for HP⁻ PUD) and one HP⁻ GU locus near *ZNF169* (rs12347577; Cochran's *Q* test, $P_{het} < 0.05$). On the other hand, the lead variants in the most significant locus at *PSCA* did not show significant heterogeneity in effect sizes ($P_{het} < 0.05$). Colocalization analysis suggested that HP⁺ PUD and HP⁻ PUD shared the causal variant in *PSCA* (PP4 > 0.8 for PUD; Supplementary Table 25).

## Genetic analyses revealed heterogeneity of GU

To further explore the similarities and differences of genetic architecture between GU and DU, we first compared the effect sizes of distinct signals identified in east Asians (lead variants and independent secondary variants) for GU and DU (Fig. 5c and Supplementary Fig. 18). Notably, the effect sizes for GU showed a strong correlation with those for DU (29 variants with MAF > 0.01, r = 0.92), which was concordant with the high genetic correlations described above. However, the effect sizes for GU were systematically smaller than those for DU (intercept = 0.01, slope = 0.43 and s.e.m._slope = 0.03), with 19 variants (19 of 29 = 65.5%) showing significant heterogeneity ($P_{het} < 0.05$) in Cochran's *Q* test (Supplementary Table 26). To further verify the findings and avoid potential biases for the comparisons, we compared (1) the effect sizes of distinct signals of GWASs in TMM-50K, FinnGen[15] and UKB[12], (2) the effect sizes generated by excluding BU samples in the association tests in BBJ1-180K (that is, no common case in the comparison) and (3) TMM-50K-derived statistics with BBJ1-180K-derived statistics (that is, no common control in the comparison). In any of these comparisons, GU showed high correlation (for variants with MAF > 0.01, r = 0.75–0.90) with DU, although with smaller effect sizes than DU (Supplementary Figs. 19–21 and Supplementary Tables 27–29). Furthermore, we utilized SBayesS[32] to estimate the SNP heritability and polygenicity (defined as the proportion of SNPs with nonzero effects) using HapMap3 (ref. 33) SNPs from EAS-specific summary statistics (Methods). Approximately 0.22% of the variants were estimated to have nonzero effects for PUD. Compared with DU, GU showed moderately higher polygenicity but

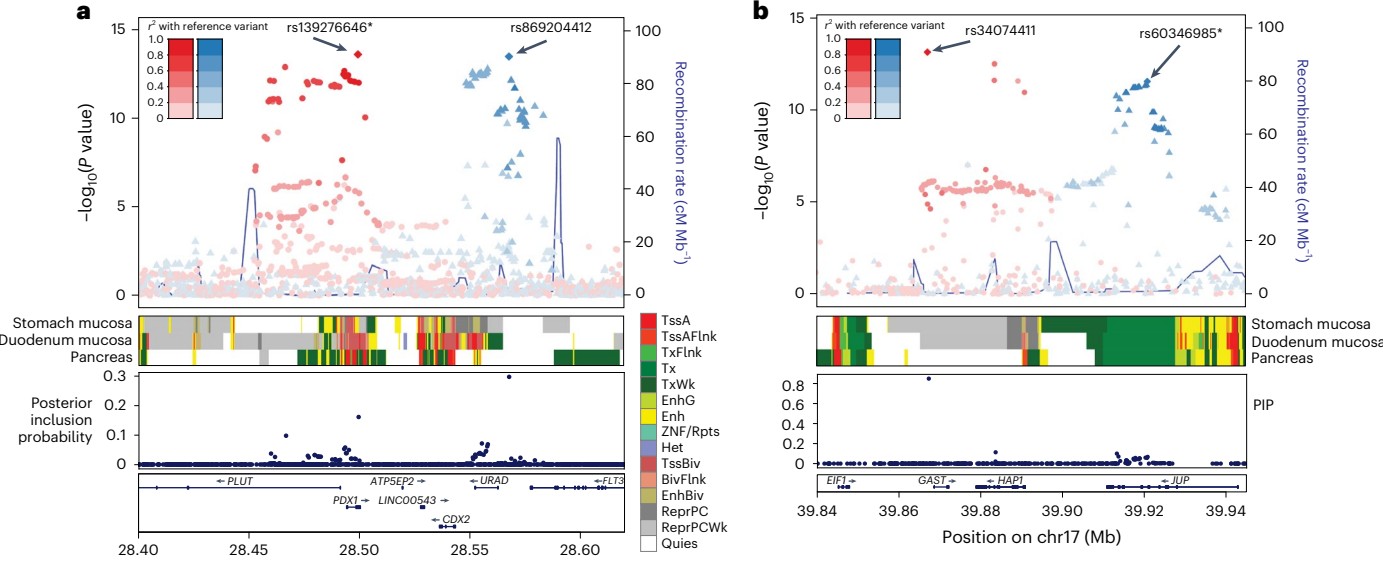

**Fig. 4 | EAS-specific secondary signals at PDX1 and JUP. a**, Regional plot at *PDX1–CDX2* locus for PUD association from east Asian-specific meta-analysis. Variants are colored to match the lead SNPs in the highest LD and the extent of LD with the lead variant is shown by a color gradient (red or blue). PIP was derived from fine-mapping analysis. Variants are plotted against GRCh37 (hg19). Chromatin states (core 15-state model) are shown for three related tissue types, namely stomach mucosa, duodenal mucosa and pancreas. BivFlnk, flanking

bivalent TSS/Enh; Enh, enhancers; EnhBiv, bivalent enhancer; EnhG, genic enhancers; Het, heterochromatin; ReprPC, repressed polycomb; ReprPCWk, weak repressed PolyComb; Quies, quiescent/low; TssA, active transcription start site (TSS); TssAFlnk, flanking active TSS; TssBiv, bivalent/poised TSS; Tx, strong transcription; TxFlnk, transcription at gene 5′- and 3′-ends; TxWk, weak transcription; ZNF/Rpts, ZNF genes & repeats. **b**, Regional plot at *GAST-JUP* locus for PUD association from east Asian-specific meta-analysis.

lower heritability (Pi$_{GU}$ = 0.24%, Pi$_{DU}$ = 0.10%; Supplementary Figs. 22–23 and Supplementary Table 30). The results demonstrated that GU and DU showed a high genetic correlation with most risk loci shared and suggested higher heterogeneity of GU[34].

Finally, using the summary statistics derived from TMM-50K, we generated polygenic risk score (PRS) models with PRS-CS[35] comprising 1,029,637 variants and tested the PRS in BBJ1-180K for associations with PUD or PUD subtypes to investigate the genetic overlap among PUD and PUD subtypes. Compared with HP+ PRS, HP− PRS generally showed stronger associations with PUD or PUD subtypes. The strongest association was between DU PRS and DU (OR = 1.22, 95% CI = 1.20–1.25, Δ$R^2$ = 0.94%). DU PRS (OR = 1.08, 95% CI = 1.06–1.10, Δ$R^2$ = 0.13%) showed a stronger association with GU than GU PRS (OR = 1.04, 95% CI = 1.03–1.06, Δ$R^2$ = 0.05%; Supplementary Table 31). The results further validated that GU shares risk loci with DU while having higher heterogeneity than the latter.

### Pleiotropy of PUD risk variants on GC

Considering that DU appears to be a protective factor against GC, we conducted two-sample Mendelian randomization (MR)[36] to evaluate the causality of PUD or its subtypes on GC. Summary statistics for GC in EAS were obtained from a previous study in BBJ1-180K, which included approximately 6,500 cases[30] (Methods). Summary statistics for PUD and its subtypes were obtained by conducting a meta-analysis combing three replication datasets. Although PUD and its subtypes showed significant ($P$ < 0.05/15, Bonferroni's correction) protective effects against GC using the IVW method (Supplementary Table 32), MR-Egger analysis suggested significant pleiotropy for the instruments (Supplementary Table 33). MR-PRESSO[37] was utilized to correct for the horizontal pleiotropic variants (ranging from six to seven for each exposure). The outlier-corrected MR showed no significant effects of PUD or its subtypes on GC (Supplementary Table 34). We note that splitting samples and removing outliers may cause power loss (Supplementary Fig. 24).

To evaluate the pleiotropic effects of PUD risk variants on GC risk, we compared the effect sizes of distinct signals in EAS between PUD subtypes and GC. For 23 available variants existing in both datasets, we found the effect sizes for DU to be negatively correlated with that for GC (slope = −0.33, s.e.m.$_{slope}$ = 0.10; Fig. 5d, Supplementary Table 35 and Supplementary Fig. 25). It was noteworthy that lead variants linked to *EFNA1* (encoding Ephrin A1, a member of the EFN family), *PTGER4* (encoding the receptor for prostaglandin E$_2$) and *PSCA* showed relatively strong but opposite effects on DU and GC (Supplementary Table 35). This suggested that the alleles of these variants, which increased the risk for PUD, could decrease the risk for GC. When removing the three variants from the regression, negative correlation was not observed for the 20 variants (slope = −0.06, s.e.m.$_{slope}$ = 0.10; Supplementary Fig. 25), indicating that negative correlation between DU and GC was largely affected by the three variants.

### Gene-based and gene-set analysis

Gene-level analysis using MAGMA[38,39] (Methods) detected 29 genes significantly associated with PUD ($P$ < 6.5 × 10$^{-7}$; Supplementary Table 36), 45 genes associated with DU and 15 genes associated with GU, in the EAS population. In total, 47 distinct genes were associated with PUD or PUD subtypes. Multiple genes identified in gene-level analysis were reportedly related to GC (*PTGER4* (ref. 40), *PRKAA1* (ref. 41), *GNAQ*[42], *GNAS*[43], *NHEJ1* (ref. 44), *IHH*[45] and *JUP*[46]). Based on the gene-level statistics, we additionally performed pathway enrichment analysis and identified one gene set after Bonferroni's correction (nikolsky_breast_cancer_8q23_q24_amplicon, including genes within amplicon 8q23-q24 identified in a study of breast tumors[47]; $P$ < 8.0 × 10$^{-7}$; Supplementary Table 37).

### Tissue- and cell-type specificity analysis

We tested the tissue-level specificity employing MAGMA[39] with GTEx v.8 datasets[21] in EAS individuals to investigate the tissue types related to PUD and its subtypes. Significant genetic enrichments (FDR < 5%)

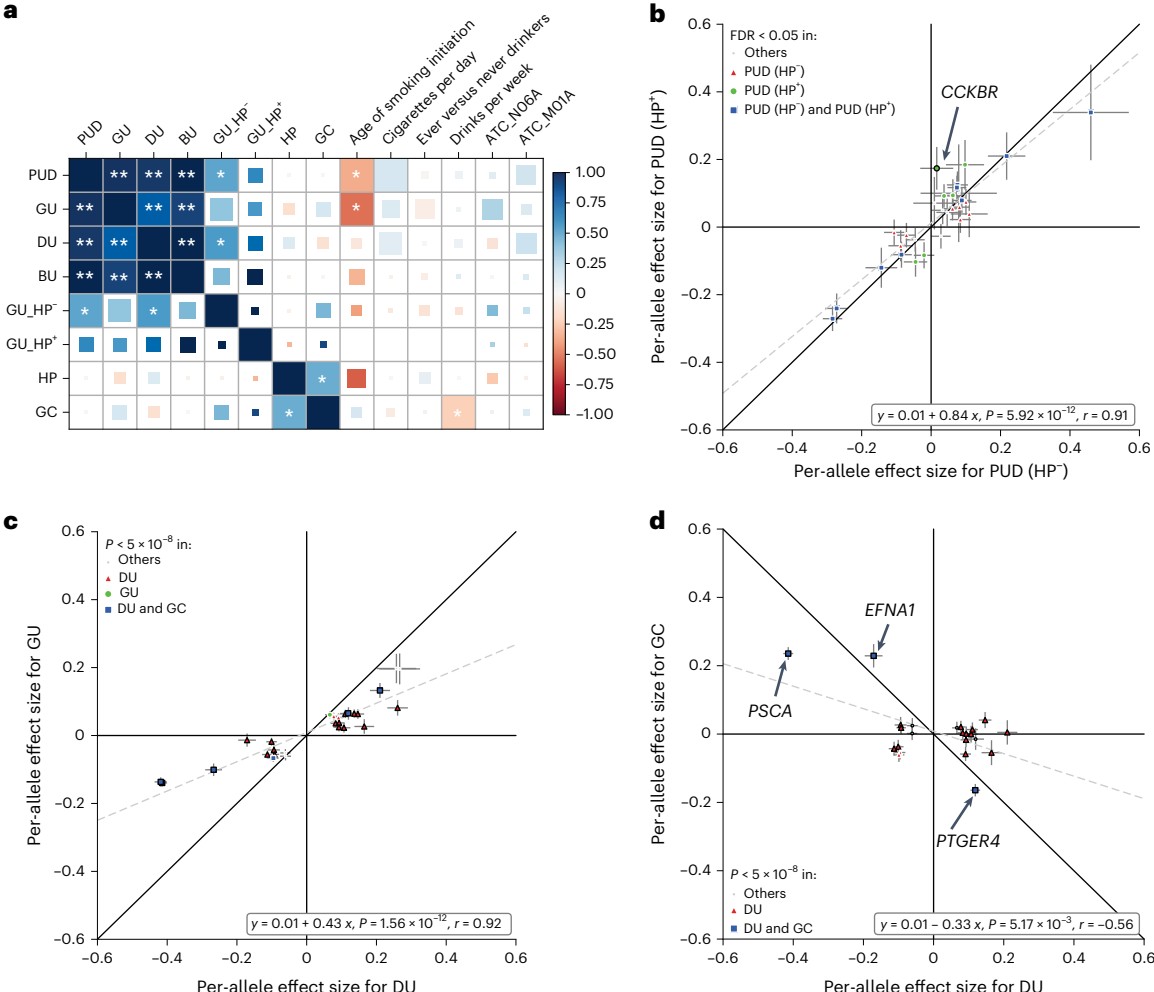

**Fig. 5 | Effect-size comparison of distinct variants and genetic correlations across PUD-related traits in the east Asian population. a**, Genetic correlation among PUD, PUD-related phenotypes and risk factors. *Two-sided $P < 0.05$; **FDR < 5%. The square size in each cell is proportional to $-\log_{10}(P)$. HP infection status determined by anti-HP IgG level; GU_HP$^+$, HP$^+$ GU; GU_HP$^-$, HP$^-$ GU. **b**, Effect-size comparison for PUD using summary statistics from HP-stratified analysis. PUD(HP$^+$), HP$^+$ PUD; PUD(HP$^-$), HP$^-$ PUD. **c**, Per-allele effect-size (log(OR)) comparison using EAS-specific summary statistics for DU and GU. Lead variants and secondary signals associated with PUD or any subtype in the EAS population were selected for comparison (GWAS $P$ values are two sided). The

most significant associations were shown if overlapping variants existed (interval <500 kb). Only variants with MAF > 0.01 are shown. Black marker edges denote variants with nominally significant heterogeneity (Cochran's $Q$ test; two-sided $P_{het} < 0.05$). The gray dashed line represents the fitted linear regression line with annotation at the bottom right ($P$ values are derived from two-sided Student's $t$-tests for the slopes). Pearson's $r$ is shown. **d**, Effect-size comparison between DU and GC. Effect sizes for DU were obtained from the EAS-specific meta-analysis. GC summary statistics were obtained from previous GWASs conducted in BBJ1. In **b**–**d**, data are presented as effect-size estimates (log(OR)) ± s.e.m.

were observed in the stomach, pancreas, small intestine and kidney for PUD, in the stomach, pancreas and prostate for DU and in the stomach for GU (Fig. 6a and Supplementary Table 38).

To further characterize specific cell types associated with PUD in the gastric and duodenal tissues, we utilized publicly available single-cell RNA sequencing (scRNA-seq) datasets of the human stomach and duodenum[48]. We performed cell-specificity analysis using LDSC[49] and MAGMA in EAS and EUR individuals, respectively (Methods). To increase statistical power, we conducted a fixed-effect meta-analysis combining EAS and EUR results for each method (Methods and Supplementary Figs. 26–31). For PUD, we found that stomach D cells reached the significance threshold (FDR < 5%) in the analyses of both MAGMA and LDSC (Fig. 6b,c and Supplementary Tables 39 and 40). In addition, duodenal enterochromaffin cells (EC cells), stomach antral EC cells and stomach tuft cells were significantly (FDR < 5%) associated with PUD, as per MAGMA (Fig. 6b). Somatostatin produced by stomach D cells inhibits the secretion of a variety of gastrointestinal

hormones, including the gastrin secreted by stomach G cells which stimulates gastric acid secretion. EC cells secrete serotonin (5-HT, a neurotransmitter) with diverse gastrointestinal functions and tuft cells (chemosensory epithelial cells) secrete interleukin-25, driving the type 2 immune response to parasitic infection. Together, the findings suggested the important role of gastrointestinal hormone regulation and immune response in PUD etiology.

## Discussion

Our GWAS meta-analyses of PUD and PUD subtypes discovered 33 autosomal susceptibility loci, of which 25 had not been reported in previous GWASs (19 in east Asian-specific analysis and 6 in cross-ancestry analysis). The loci were mostly shared across ancestries with strong correlations of effect sizes. Our cross-ancestry analysis emphasized the high genetic correlation of DU and suggested the heterogeneity of GU across ancestries. The larger effect sizes of *MUC1* and *MUC6* in populations of EUR ancestry are in reasonable agreement with their

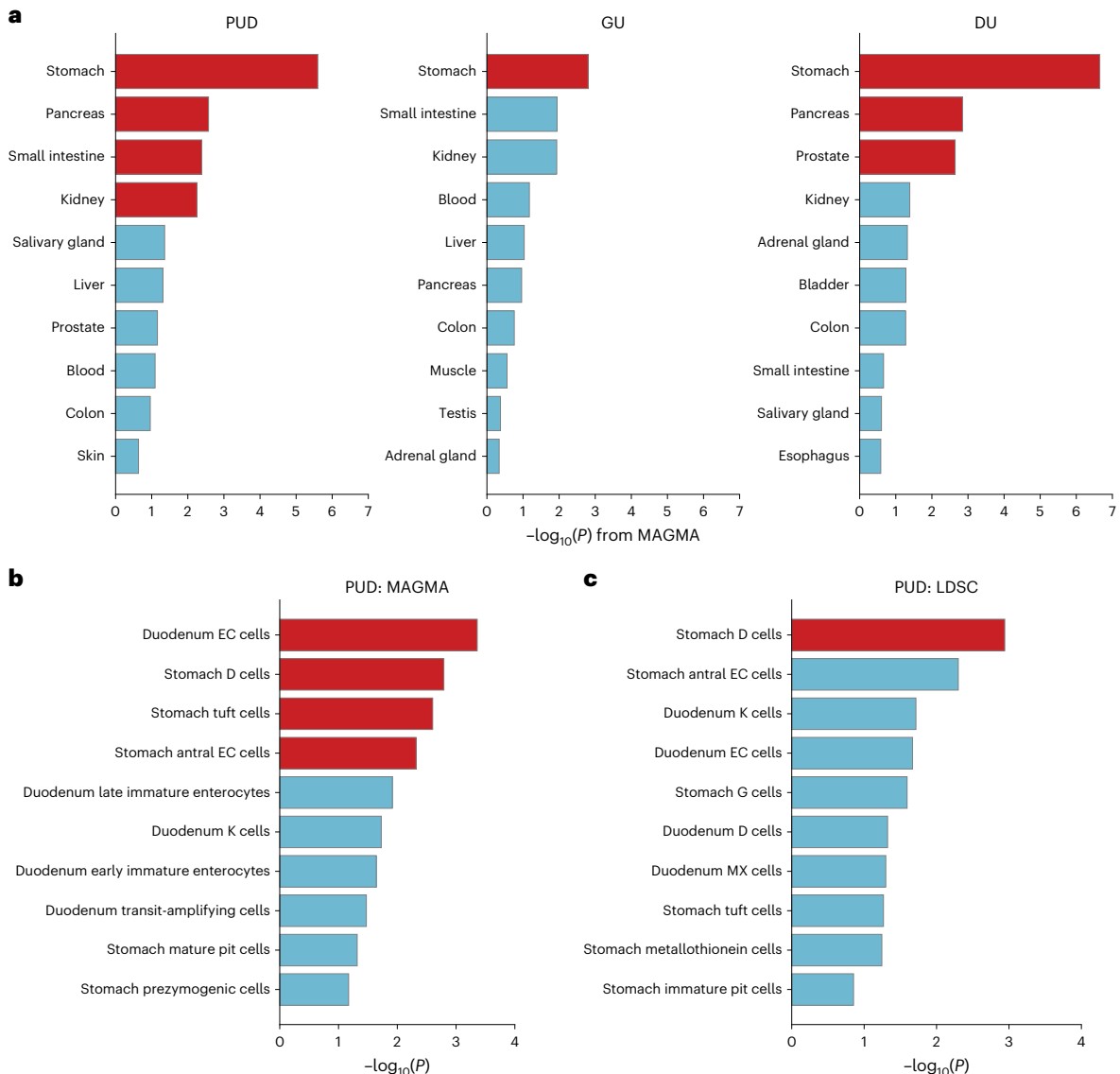

**Fig. 6 | Tissue- and cell-type specificity analysis. a**, Associations across PU phenotypes and 30 general tissue types analyzed using MAGMA with east Asian-specific summary statistics and the GTEx v.8 dataset. The $-\log_{10}(P)$ (one-sided $P$ values) are shown. **b,c**, Associations between PUD and cell types in the stomach and duodenum analyzed using MAGMA (**b**) and LDSC (**c**) (testing for enrichment of the 10% most specific genes in each cell type). IVW meta-analysis combined statistics from east Asian and European ancestries for each method. The $x$ axis shows $-\log_{10}(P)$ (one-sided $P$ values) derived from meta-analyzed estimates. In **a**, **b** and **c**, the red bars imply significant associations after corrections for each analysis (FDR < 5%). Only the ten most significant associations for each phenotype are shown.

critical roles in the protection from NSAID-induced injury, given the much lower prevalence of HP infection in western countries compared with that in east Asian populations[50].

Multiple new loci (*PAX4* (ref. [51]), *PDX1* (ref. [52]), *IHH*[53] and *SLC22A3* (ref. [54])) and reported loci (*CCKBR*, *CDX2* and *GAST*) were found to be related to cell differentiation or gastrin signaling. By integrating scRNA-seq datasets, we identified the association of PUD with certain hormone-secreting cells, including stomach D cells (somatostatin) and stomach antral and duodenal EC cells (5-HT). The potential roles of D cells and EC cells are discussed in Supplementary Note. Our results also showed the signal at *CCKBR* (encoding the receptor for gastrin) to be HP$^+$ specific. The PUD risk allele of the lead SNP (rs12792379) is in LD with the eQTL allele associated with higher *CCKBR* expression in multiple tissues[21], including esophagus mucosa. It has been widely shown that HP-elicited cytokines stimulate gastrin release[55]. It is likely that the increased gastrin level induced by HP will interact with altered expression in *CCKBR*, leading to dysregulated gastric acid secretion and altered susceptibility to apoptosis[56,57]. Taken together, our results provided genetic evidence of gastrointestinal cell differentiation and hormone regulation being critical in PUD etiology.

As expected, we observed high genetic correlation between GU and DU and nominally significant genetic correlations between PUD and its risk factors (Supplementary Note); effect-size comparisons demonstrated that GU shared risk loci with DU, but had smaller effect sizes than DU. Polygenicity of GU was higher than that of DU. SNP-based heritability estimate for DU (liability scaled) was almost twice as high as for GU. In addition, DU PRS showed a stronger association with GU than GU PRS in east Asians. The results revealed the genetic difference between GU and DU and reflected a higher heterogeneity of GU[58–60].

We found three variants (linked to *EFNA1*, *PTGER4* and *PSCA*) to have relatively large pleiotropic effects on DU and GC. EFNA1 suppresses tumor growth whereas PGE2 supports tumor growth by

promoting angiogenesis[40,61]. PUD risk alleles resulted in increased levels of EFNA1 and reduced levels of *PTGER4*, whereas GC risk alleles were associated with a decreased level of EFNA1 and an increased level of *PTGER4*. This suggested that the risk alleles of variants at *EFNA1* and *PTGER4* for GC (nonrisk alleles for PUD) potentially benefited peptic ulcer healing while imposing an increased risk for GC through upregulated cell proliferation and angiogenesis. In addition, we also detected multiple PUD risk, cancer-related genes (for example, *IHH*, *GNAS*, *NHEJ1*, *JUP* and *MECOM*), which provided potential targets contributing to the different outcomes of PUD or GC. No causal effects were identified in the outlier-corrected MR in the present study, which may suffer from the power loss caused by sample split and removal of variants. Further research is warranted on the protective role of DU against GC.

Although we identified multiple associations, the present study has several potential limitations. First, the phenotypic information of PUD and subtypes was obtained via interviews and reviews of medical records. However, the prevalence rate of PUD was consistent with that in previous epidemiological studies; our study replicated most of the previously identified loci and the new biological findings are feasible, which suggested the relatively high reliability of the results. Second, due to the lack of information about the chronological order of PUD onset and anti-inflammatory drug use at PUD onset in the present study, the specific interaction of NSAIDs with host genetic factors was underexplored. Third, detailed information on the anatomical site of the ulcers or the strains of HP was not available. Fourth, the variants identified by fine-mapping and the overlaps in association signals identified by lookup approaches might result from tagging distinct causal variants, and the meta-analysis fine-mapping using BBJ1-180K as LD reference might be miscalibrated[62], which should be interpreted cautiously. Despite the Biobank-scale cohort for HP-stratified analysis, the statistical power is still limited for certain analyses. Even though our subtype analysis revealed the overall similarities and differences in genetic architecture, a large sample size and more detailed classifications are still warranted to elucidate the potential heterogeneity further.

In summary, the present study approximately quadrupled the number of risk loci for PUD and its subtypes and improved our understanding of the genetic architecture of PUD. The findings provided insight into the biological pathways involved in PUD pathogenesis and potential links between PUD and GC. We demonstrated that, besides HP-related loci, host genetic factors potentially involved in gastric hormone regulation, cell differentiation and proliferation might play important roles in PUD pathogenesis. Our single-cell analysis further revealed the association of 5-HT-secreting EC cells, somatostatin-secreting stomach D cells and stomach tuft cells with PUD, indicating their key role in PUD etiology.

## Online content

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

## BioBank Japan

**Koichi Matsuda[5], Takayuki Morisaki[4,5], Yoichiro Kamatani[1], Yoishinori Murakami[4] & Akiko Nagai[6]**

A full list of members and their affiliations appears in the Supplementary Information.

## Methods

### Study participants

We included four cohorts for east Asian-specific meta-analysis including BBJ1-180K, BBJ1-12K and BBJ2-42K from the BioBank Japan Project[10], and TMM-50K from Tohoku University Tohoku Medical Megabank[63]. Further details on each cohort are described in Supplementary Note. Sample overlap was checked in BBJ1-180K, BBJ1-12K and BBJ2-42K by merging three datasets and estimating the identical-by-descent sharing, and the number of potentially overlapping samples is minimal (<0.06%; Supplementary Fig. 32).

The clinical characteristics of these cohorts are provided in detail in Supplementary Table 1. The research project was approved by the ethics committees at the Institute of Medical Science, the University of Tokyo (application no. 29-74-A0215) and Iwate Tohoku Medical Megabank Organization, Iwate Medical University (application no. HG H25-2).

### Phenotype definition

In the present study, we assessed PUD, which is a combination of the two major subtypes, namely DU and GU. Cases with PUD were obtained from the combination of individuals with any of the two major PUD subtypes (DU and GU). Clinical information for cases with GU and DU was obtained via interviews and reviews of medical records using a standardized questionnaire in BBJ1-180K, BBJ1-12K and BBJ2-42K, and by self-administered questionnaires in TMM-50K[63].

Individuals with comorbidities of GU and DU were additionally categorized into group BU. Individuals without a given diagnosis of peptic ulcers or any HP-related diseases (GU, DU or GC) were used as control samples (Supplementary Table 1). The cohorts and phenotypes that were first reported in the present study have been summarized in Supplementary Table 41.

### Genotyping and imputation

All samples included in the east Asian-specific analysis are genotyped using commercially available genotyping arrays. QC of autosomal genotypes was performed as described previously. Detailed information on genotyping arrays and QC procedures are provided in Supplementary Note[64].

Pre-phasing was conducted using Eagle2 (v.2.4.1; https://alkes-group.broadinstitute.org/Eagle)[65]. Imputation was performed with Minimac4 (v.1.0.2; https://github.com/statgen/Minimac4) using the 1000 Genomes Project Phase 3 (ref. 11) v.5 (1KGp3v5) ALL panel (https://genome.sph.umich.edu/wiki/Minimac3#Reference_Panels_for_Download). BBJ1-12K and BBJ2-42K were additionally imputed with the 1KG high-coverage reference panel[66] (GRCh38). For chromosome X, haplotypes were pre-phased for men and women, and variants were imputed separately in men and women using the same software. Imputed variants with $R^2 < 0.3$ were excluded in the association analysis. More than 12 million variants were included in the discovery-stage association analysis.

### Genome-wide association analysis

Single-variant association analysis was performed with SAIGE (v.0.44; https://github.com/weizhouUMICH/SAIGE)[12], which implements a generalized mixed model with SPA correction controlling for case–control imbalance and cryptic relatedness. The regression included age, sex and the top ten PCs as covariates. For step 1, LD-pruned genotyped variants (PLINK–indep-pairwise 50 5 0.2, https://www.cog-genomics.org/plink)[67] with MAF > 1% were used to estimate the null models with leave one chromosome out (LOCO). Variants with MAC < 20 were excluded from the association tests (step 2). For sex-stratified analysis, single-variant association analyses were performed with SAIGE, adjusting for the same set of covariates other than sex in men and women.

For association tests of the X chromosome, variants were tested separately in men and women using the corresponding null models estimated by autosomes in each sex. Haploid-based dosages of the nonpseudo autosomal region of men were multiplied by 2. The results for each sex were then meta-analyzed using IVW methods implemented in METAL software (v.2011-03-25, http://csg.sph.umich.edu/abecasis/Metal/index.html)[68].

Genome-wide significant loci were determined by iteratively extending 500-kb flanking regions around the most significant variant until no genome-wide significant variant ($P < 5.0 \times 10^{-8}$) was detected within the extended regions. The most significant variant in each locus was selected as the lead variant. Loci for different traits with lead variants within 500 kb of each other were considered the same, denoted by the most significant lead variant from the locus. Significant variants in the major histocompatibility complex (MHC) region (GRCh37, chromosome 6: 25–34 Mb) were counted as one locus due to the complexity of the region.

### LDSC

We performed LDSC (v.1.0.0; https://github.com/bulik/ldsc)[14] to examine the bias caused by confounding factors, such as population stratification or cryptic relatedness. We employed the LD scores provided by authors for the east Asian population, which were estimated from 1KG EAS individuals. To convert observed-scale heritability to liability-scale heritability, the population prevalence rates in east Asian populations were set to 6.2%, 6.9%, 10.8% and 1.8% for DU, GU, PUD and BU, respectively. The prevalence rates were estimated from the population-based TMM-50K and were similar to those in previous epidemiological studies[1].

### Replication of significant associations and EAS-specific meta-analysis

We compared the directions and effect sizes with the replication GWAS sets for the lead variants of significant loci identified in the discovery-stage GWASs. The results of GWASs at the discovery and replication stages were combined using the fixed-effect, inverse-variance method implemented in METAL. Heterogeneity was estimated by Cochran's $Q$ test. In addition, the random-effects model implemented in GWAMA[69] was employed to evaluate the heterogeneity (Supplementary Note and Supplementary Fig. 33). We considered the lead variants identified in discovery-stage GWASs as replicated if the variants reached a nominal significance threshold ($P_{rep} < 0.05$) in the same direction in at least two of the replication GWASs.

### Cross-ancestry meta-analysis

Summary statistics of PUD and PUD subtypes for European individuals were obtained from FinnGen (release 6 for PUD, DU and GU; https://www.finngen.fi/en/access_results)[15], a published GWAS of PUD in UKB (https://cnsgenomics.com/content/data)[5] and PheWeb UKB-SAIGE[12] (DU and GU; https://pheweb.org/UKB-SAIGE) (details in Supplementary Table 7).

Genome coordinates of summary statistics were converted from GRCh38 (hg38) to GRCh37 (hg19) using the University of California Santa Cruz LiftOver tool[70] if the original summary statistics were based on GRCh38. We performed additional QC, variant normalization[71] and harmonization for all summary statistics before meta-analyses. Details of QC and harmonization are presented in Supplementary Note.

The fixed-effect, inverse-variance method was used to conduct meta-analyses integrating GWAS results in EAS and EUR populations using METAL. We additionally performed fixed-effect meta-analyses to generate EUR-specific summary statistics using the two EUR datasets. Population-specific meta-analyses were used to compare the effect sizes of lead variants identified in cross-ancestry meta-analyses between EAS and EUR populations. To investigate potential associations in GRCh38-specific regions, we further conducted a cross-ancestry meta-analysis combining GRCh38-based datasets (Supplementary Note and Supplementary Figs. 34–35).

MR-MEGA (v.0.2: http://www.geenivaramu.ee/en/tools/mr-mega)[16] was used to perform cross-ancestry meta-regression with four axes of genetic variation derived via multidimensional scaling. $P$ values were recalculated using the $\chi^2$ statistic due to the lack of support in MR-MEGA for $P < 1.0 \times 10^{-14}$.

## Genetic correlation estimation

To assess the genetic correlation between PUD and common binary traits and quantitative traits in east Asian populations, we used cross-trait LDSC[28] with LD scores estimated from 1KG EAS individuals. East Asian summary statistics were obtained from previous GWASs in BBJ[29,30]. The MHC region was excluded.

To evaluate the cross-ancestry correlations of genetic effect for PUD and subtypes between EAS and EUR, Popcorn (v.1.0: https://github.com/brielin/Popcorn)[17] was used with pre-computed cross-ancestry LD scores estimated from 1KG EUR and EAS populations. For these analyses, meta-analyzed summary statistics of PUD in EAS and EUR for HapMap3 SNPs (without the MHC region) were used.

## Blood group and secretor status interaction analysis

ABO blood groups for unrelated individuals used in discovery-stage GWASs in BBJ1-180K (KING kinship coefficient[72] <0.0884, $n = 164,613$) were inferred from two genotyped variants described previously[2]. Secretor status was inferred using the best-guess genotype of imputed variants rs1047781 (p.Ile140Phe)[73], where AA or AT genotypes are secretors and TT are nonsecretors. Logistic regression was performed to examine (1) the association of blood group or secretor status with PUD and (2) blood group O–secretor status interaction. Details on the logistic regression are described in Supplementary Note.

## Conditional analysis by COJO

GCTA-COJO (v.1.93.2; https://yanglab.westlake.edu.cn/software/gcta/#COJO)[18] was employed to perform conditional analysis in each significant locus identified in EAS-specific meta-analysis of PUD and subtypes. We constructed an LD reference panel using the best-guess imputed genotype of 20,000 randomly selected and unrelated individuals of east Asian ancestry from BBJ1-180K. Stepwise model selection was conducted first to select independent association signals ($P < 5.0 \times 10^{-8}$) and a joint analysis of these selected signals was performed next. Variants with MAF > 0.01 were included in the analysis.

## Fine-mapping and variant annotation

Fine-mapping was conducted using SuSiE (v.0.11.92; https://github.com/stephenslab/susieR)[19] with default configurations while allowing ten putative causal variants within each locus. Unrelated individuals (KING kinship coefficient <0.0884, $n = 171,085$) from BBJ1-180K were used as LD reference, computed by LDstore (v.2.0; http://www.christianbenner.com)[74] based on the imputed dosages.

We defined regions based on the 3-Mb window centered at the lead variants and merged them if the window overlapped. Only variants with $R^2 \geq 0.5$ were included in fine-mapping. We reported the missense variants in credible sets, which have a 95% probability of harboring one causal variant.

Variants identified in the GWASs were annotated using ANNOVAR (v.2020-06-07; -protocol refGene,avsnp150,clinvar_20200316; https://annovar.openbioinformatics.org/en/latest)[75]. Chromatin states (core 15-state model) for stomach mucosa and duodenum mucosa were obtained from the Roadmap Epigenomics Project[76]. Allele frequencies for variants not available in population-specific meta-analysis were obtained from the Genome Aggregation Database[77] (https://gnomad.broadinstitute.org) or the 1KG. LocusZoom[78] was used to create the region plot.

## HP-stratified analysis

To investigate the interaction of HP with host genetic factors for the development of peptic ulcers, we performed HP-stratified analyses in HP+ and HP− individuals from TMM-50K. HP infection status was determined by anti-HP serum immunoglobulin (Ig)G antibody, measured by the latex agglutination immunoassay. Individuals with anti-HP serum IgG antibody titer ≥10 U ml$^{-1}$ were categorized as HP+. Association tests were performed with the same settings as in the discovery-stage GWASs. Cochran's $Q$ and $I^2$ statistics (calculated by R package metafor v.3.4: https://www.metafor-project.org/doku.php)[79] were used to test the effect-size heterogeneity between HP+ and HP− GWASs for each subtype. As stratified analysis could reduce power and lead to false-negative results, we estimated the power of GWASs of HP− PUD for the identified HP+-specific variant. (Supplementary Fig. 36).

Colocalization analysis was conducted using the coloc package (v.5.1.0; https://chr1swallace.github.io/coloc)[80] for each significant locus identified in EAS meta-analysis under a single causal variant assumption. For loci with multiple independent signals identified in the conditional analysis, coloc was applied to the signals identified by SuSiE[81].

## Estimation of polygenicity using SBayesS

To estimate the polygenicity (defined as the proportion of SNPs with nonzero effects) and the strength of negative selection (defined as the relationship between MAF and effect sizes, and denoted by S) for PUD, we utilized SBayesS from GCTB software (v.2.0: https://cnsgenomics.com/software/gctb/#Overview)[32]. SBayesS employs a Bayesian mixed linear model and reports the posterior means of SNP-based heritability, polygenicity estimates and a metric that indicates negative selection.

An LD reference panel for EAS was constructed using the approach described previously[32]. For a crosstrait comparison of polygenicity estimates in EAS, we included 42 binary traits of BBJ1 in the analysis[35]. Details of the construction of the LD reference and parameters for SBayesS are provided in Supplementary Note[30].

## PRS construction and evaluation

PRS models for PUD and PUD subtypes were constructed in east Asians. We used the summary statistics derived from replication GWASs and HP-stratified analysis in the population-based Japanese cohort TMM-50K. PRS-CS (v.2021-Jun-4, https://github.com/getian107/PRScs)[35] and Python (v.3.8.8) were employed to compute PRS models using HapMap3 SNPs with an EAS-specific LD reference panel from the 1KG. Global shrinkage parameters were obtained from the data by PRS-CS using a fully Bayesian approach (PRS-CS-auto). We applied the models in BBJ1-180K and then tested the associations of PRS with PUD and PUD subtypes using logistic regression adjusted for age, sex and the top five PCs. We evaluated the predictive ability of each PRS model by its improvement of the area under the curve and $R^2$ on the liability scale[82] over a base model that includes age, sex and the top five PCs.

## Two-sample MR analysis

MR analysis was performed using TwoSampleMR[36] to evaluate the causality of PUD or its subtypes on GC. To avoid sample overlap between the exposure and outcome, we conducted an additional meta-analysis combing GWASs for PUD and its subtypes in BBJ1-12K, BBJ2-42K and TMM-50K. A total of 23 available independent variants identified in the EAS-specific meta-analysis were used as instrumental variables. To avoid bias caused by weak instruments, we further estimated per-variant $F$ statistics for each exposure and removed variants with $F < 10$ from the instrumental variables for the exposure. MR-PRESSO[37] was employed to correct for the horizontal pleiotropic variants. The summary statistics for GC were obtained from the previous study conducted in BBJ1-180K[30]. Statistical power for MR was approximately estimated using methods described previously[83]. Compared with MR

using all available variants, the pleiotropic outlier correction may result in insufficient power (Supplementary Fig. 24).

### Gene-based analysis and pathway analyses

Gene-based and pathway analyses were performed using MAGMA (v.1.08: https://ctg.cncr.nl/software/magma)[38] implemented in FUMA (v.1.3.8: https://fuma.ctglab.nl)[39]. An LD reference panel constructed from 1KG EAS population was used. A total of 19,033 protein-coding genes (ENSEMBL[84] v.92) were tested. The results of the gene-based analysis were then employed to conduct gene-set enrichment analysis with a total of 15,485 curated gene sets and gene ontology terms from MsigDB[65] v.7.0. We conducted pairwise comparisons of the $-\log_{10}(P)$ values generated by MAGMA, using a range of window sizes to evaluate the robustness of the association results in gene-based, tissue-type specificity and cell-type specificity analyses discovered by MAGMA; the effect of window size selection was marginal (Supplementary Figs. 37–39).

### Per-allele effect-size comparison

For pairwise effect-size (logarithm of ORs) comparison among PUD, PUD subtypes and GCs in the EAS population, we selected the nonoverlapping (interval between adjacent variants >500 kb) lead variants identified by EAS-specific meta-analysis and the independent signals identified by COJO analysis. For loci associated with two or more phenotypes, we selected the most significant associations (lead variants with the lowest $P$ value) for comparison. Effect sizes in EAS-specific meta-analysis were used for PUD and PUD subtypes. Summary statistics for GCs were obtained from a previous study in BBJ1-180K[13]. For cross-ancestry comparison of variant effect sizes, we included all nonoverlapping lead variants associated with PUD or any subtypes in population-specific meta-analysis or cross-ancestry meta-analysis. Associations, with the lowest $P$ values, of loci associated with more than one phenotype were selected for comparison and effect sizes in the population-specific meta-analysis were used. Cochran's $Q$ test was used to test heterogeneity across the effect sizes. In addition, we compared the WC-corrected effect sizes of the lead variants identified in the EAS meta-analysis in the present study and the variants reported in UKB[5] using the methods described previously[85,86].

### PheWAS in BBJ

To investigate whether variants associated with PUD were also associated with other human complex traits in EAS, statistics of the nonoverlapping lead variants and secondary signals for 215 case–control and quantitative traits were obtained from BBJ PheWeb (https://pheweb.jp)[7]. The LD proxy (LD $r^2 > 0.6$) with the highest $r^2$ estimated from 1KG EAS was used if a variant was unavailable in the datasets. After multiple-test corrections, the significance threshold was set to $P < 8.6 \times 10^{-6}$.

### Tissue- and cell-type specificity analysis

MAGMA[38] gene-property analysis implemented in the SNP2GENE method of FUMA[39] was employed for tissue-type specificity analysis with gene expression profile from the GTEx v.8 dataset[21]. A total of 54 nondiseased tissue types and 30 general tissue types were tested. Tissue types with an FDR < 5% were considered significant.

To identify the cell types associated with PUD in the stomach and duodenum, processed scRNA-seq datasets of the human stomach and duodenum were obtained from a previous study[48], which filtered for cells with >1,500 transcripts per cell and genes expressed by at least three transcripts in at least one cell. A total of 13,980 genes for 19 cell types in the stomach and 17 cell types in the duodenum were included in the study. The top 10% most specifically expressed genes based on fold-change (defined as the average transcript counts of all cell types except the target cell type divided by the average transcript counts of the target cell type) were extracted for each cell type. SNPs in

cell-type-specific genes were used to compute partitioned LD scores in the 1KG Phase 3 EAS or EUR population. The gene coordinates were extended by a window size of 100 kb to capture the effects of regulatory elements. Stratified LDSCs[49] were performed using the partitioned LD scores of cell-type-specific genes, partitioned LD scores of all available genes in the dataset and the baseline model of 53 annotations for each ancestry on HapMap3 SNPs, excluding the MHC region (downloaded from https://alkesgroup.broadinstitute.org/LDSCORE).

We performed gene-set enrichment analysis using MAGMA with the cell-type-specific gene sets described above. We used 1KG Phase 3 EAS and EUR population datasets as reference panels. Variants with MAF < 0.01 or in the MHC region were excluded from the analysis. The gene coordinates were extended by window sizes of 35 kb upstream and 10 kb downstream. IVW meta-analysis was performed using statistics of both ancestries for each method to increase statistical power. $P$ values were calculated using the one-tailed test. Cell types with FDR < 5% within each expression dataset were considered significant.

### The eQTL and pQTL analyses

To characterize the effect of variants on gene expression level, we extracted LD proxies (with LD $r^2 > 0.6$) in EAS or EUR 1KG Phase 3 with the lead variants and secondary signals in EAS. We extracted only significant SNP–gene pairs with FDR < 5% (pre-computed by the authors) from GTEx v.8 (ref. 21). We checked the overlap between the lead variants and secondary signals (including proxies) and cis-eQTL variants in GTEx v.8. The most significant cis-eQTL association for each gene in each tissue was selected for interpretation.

To characterize the effect of variants on protein level, we extracted LD proxies (with LD $r^2 > 0.6$) in EAS or EUR 1KG Phase 3 with the lead variants and secondary signals in EAS. We extracted genome-wide significant SNP–protein associations from five published, large-scale pQTL studies in recent years[22,24–27], conducted in individuals of mainly European ancestry. We then checked the overlap between the lead variants (including LD proxies) with cis- and trans-pQTL. The most significant association for each protein was selected for interpretation.

### Reporting summary

Further information on research design is available in the Nature Portfolio Reporting Summary linked to this article.

## Data availability

Summary statistics for GWAS of PUD and PUD subtypes in BBJ1-180K, BBJ1-12K and BBJ2-42K, and EAS-specific and cross-ancestry meta-analysis summary statistics are available at the National Bioscience Database Center (NBDC, https://humandbs.biosciencedbc.jp) Human Database (research ID: hum0311) and Japanese ENcyclopedia of GEnetic associations by Riken (JENGER, http://jenger.riken.jp/result; case–control GWAS nos. 135–155). EAS-specific and cross-ancestry meta-analysis summary statistics are additionally deposited to the European Bioinformatics Institute GWAS catalog (https://www.ebi.ac.uk/gwas) (study accession nos. GCST90270926–GCST90270932). Summary statistics derived from TMM-50K (GWAS of PUD and PUD subtypes; HP-stratified analysis) are available at the Japanese Multi Omics Reference Panel (jMorp, https://jmorp.megabank.tohoku.ac.jp; ID: TGA000011). Genotype data for BBJ were deposited at the NBDC Human Database (BBJ1-180K, research ID: hum0014; BBJ1-12K and BBJ2-42K, research ID: hum0311). Summary statistics for European individuals were obtained from FinnGen release 6 (https://www.finngen.fi/en/access_results) and UKB datasets (https://cnsgenomics.com/content/data and https://pheweb.org/UKB-SAIGE). Summary statistics for other traits in BBJ were obtained from JENGER (http://jenger.riken.jp), BBJ PheWeb (https://pheweb.jp/phenotypes) and NBDC (research ID: hum0014).

## Code availability

Publicly available software and packages were used for bioinformatics analysis in the present study. The software and packages used in each analysis are described in Methods and Nature Portfolio Reporting Summary.

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

## Acknowledgements

We thank all the participants and investigators of BBJ, Tohoku Medical Megabank, UKB and FinnGen. We thank M. Lathrop and C. Terao for their valuable support. This research was supported by the Ministry of Education, Culture, Sports, Sciences and Technology (MEXT) of the Japanese government and the Japan Agency for Medical Research and Development (AMED) under grant nos. JP18km0605001/ JP23tm0624002 (the BioBank Japan project), JP19km0405215 (to C.T., K.M. and Y.K.), JP22zf0127009 (to K.M.) and JP223fa627011 (to K.M. and Y.K.). This work was supported by the Tohoku Medical Megabank Project (Special Account for the Reconstruction of the Great East Japan Earthquake) from MEXT and AMED (grant nos. JP15km0105004 and JP21tm0124006), including the supercomputer resource powered by the AMED research grant (no. JP20km0405001). We thank the Human Genome Center, the Institute of Medical Science and the University of Tokyo for providing the supercomputing resources used in the present study, and the Digital Research Alliance of Canada for additional computational resources.

## Author contributions

Y.K. supervised this project. C.T., K.M., Y.K., M.S. and Y.H. designed the study. Y.H., M.S., Y.S., M.K. and H.M.M contributed to bioinformatics analyses. C.T., K.M., Y.K., M.K., A.N., Y.M. and T.M. contributed to the management of data from BBJ. C.T. contributed to the phenotyping in BBJ. Y.S., Y.O.-Y., T.H. and A.S. contributed to the management and association analyses in TMM-50K. Y.H. wrote the manuscript with critical input from M.K., M.S. and Y.K. All authors provided critical revision of the manuscript.

## Competing interests

Y.K. holds stock in StaGen Co. Ltd. T.H. is a board member of Genome Analytics Japan Inc. The other authors declare no competing interests.

## Additional information

**Correspondence and requests for materials** should be addressed to Yoichiro Kamatani.

# Reporting Summary

## Statistics

For all statistical analyses, confirm that the following items are present in the figure legend, table legend, main text, or Methods section.

| n/a | Confirmed | |
|---|---|---|
| ☐ | ☒ | The exact sample size (*n*) for each experimental group/condition, given as a discrete number and unit of measurement |
| ☐ | ☒ | A statement on whether measurements were taken from distinct samples or whether the same sample was measured repeatedly |
| ☐ | ☒ | The statistical test(s) used AND whether they are one- or two-sided<br>*Only common tests should be described solely by name; describe more complex techniques in the Methods section.* |
| ☐ | ☒ | A description of all covariates tested |
| ☐ | ☒ | A description of any assumptions or corrections, such as tests of normality and adjustment for multiple comparisons |
| ☐ | ☒ | A full description of the statistical parameters including central tendency (e.g. means) or other basic estimates (e.g. regression coefficient) AND variation (e.g. standard deviation) or associated estimates of uncertainty (e.g. confidence intervals) |
| ☐ | ☒ | For null hypothesis testing, the test statistic (e.g. $F$, $t$, $r$) with confidence intervals, effect sizes, degrees of freedom and $P$ value noted<br>*Give P values as exact values whenever suitable.* |
| ☐ | ☒ | For Bayesian analysis, information on the choice of priors and Markov chain Monte Carlo settings |
| ☒ | ☐ | For hierarchical and complex designs, identification of the appropriate level for tests and full reporting of outcomes |
| ☐ | ☒ | Estimates of effect sizes (e.g. Cohen's *d*, Pearson's *r*), indicating how they were calculated |

*Our web collection on statistics for biologists contains articles on many of the points above.*

## Software and code

Policy information about availability of computer code

| Data collection | No software was used. |
|---|---|
| Data analysis | We used publicly available software for the analysis, including Plink (v1.9 and v2.0), Eagle2 (v2.4.1), Minimac4 (v1.0.2), SAIGE (v0.44), LDSC (v1.0.0), ANNOVAR (v2020-06-07), METAL (v2011-03-25), GWAMA(v2.2.2), UCSC LiftOver tool, MR-MEGA (v0.2), Popcorn (v1.0), Python(v3.8.8), R(v4.1.0), LocusZoom (v1.2), GCTA-COJO (v1.93.2), SuSiE (v0.11.92), LDstore (v2.0), metafor (v3.4), coloc (v5.1.0), GCTB (v2.0), PRScs (v2021-Jun-4), TwoSampleMR(0.5.6), MAGMA (v1.08), and FUMA (v1.3.8). |

For manuscripts utilizing custom algorithms or software that are central to the research but not yet described in published literature, software must be made available to editors and reviewers. We strongly encourage code deposition in a community repository (e.g. GitHub). See the Nature Portfolio guidelines for submitting code & software for further information.

## Data

Policy information about availability of data

All manuscripts must include a data availability statement. This statement should provide the following information, where applicable:

- Accession codes, unique identifiers, or web links for publicly available datasets
- A description of any restrictions on data availability
- For clinical datasets or third party data, please ensure that the statement adheres to our policy

Summary statistics for GWAS of PUD and PUD subtypes in BBJ1-180K, BBJ1-12K and BBJ2-42K, and EAS-specific and cross-ancestry meta-analysis summary statistics

# Human research participants

Policy information about studies involving human research participants and Sex and Gender in Research.

| | |
|---|---|
| Reporting on sex and gender | We performed sex (biological attribute)-stratified analysis in BBJ1-180K. In this analysis, sex was assigned when the self-reporting sex matched the sex imputed from X chromosome inbreeding coefficients. In total, 78,211 males and 74,967 females in BBJ1-180K were included in the sex-stratified analysis. Detailed descriptions and results were reported in Supplementary Table 1 and Supplementary Table 3. |
| Population characteristics | BioBank Japan Project (BBJ) is a hospital-based study that recruited approximately 200,000 participants (mainly of Japanese ancestry) from 2003 to 2007 and additionally recruited approximately 67,000 participants from 2013 to 2017.<br>The Tohoku Medical Megabank Project (TMM) involves two prospective cohort studies, one is a population-based adult cohort study of 80,000 participants, and the other is a birth and three-generation cohort study of 70,000 participants. In this study, we used the data from a part of the population-based cohort study.<br>The population characteristics are provided in detail in Supplementary Table 1. |
| Recruitment | BioBank Japan Project (BBJ) recruited participants at 66 hospitals (BBJ1) and 52 hospitals (BBJ2) with the support from 12 medical institutions. In each recruitment period, patients were enrolled who were diagnosed as one of 47 common diseases between 2003 and 2007 (BBJ1), or one of 38 diseases (mostly overlapped) between 2013 and 2017 (BBJ2). (https://biobankjp.org/en/index.html#01)<br>In the population-based adult cohort study of the Tohoku Medical Megabank Project (TMM), participants were recruited on a voluntary basis through the specific health checkup conducted by municipalities and also at the seven "Community Support Centers" or five "Satellites" in Miyagi and Iwate Prefectures in Japan, from 2013 to 2016. |
| Ethics oversight | All the participants from BBJ and TMM provided written informed consent. The research project was approved by the ethics committees at the Institute of Medical Science, the University of Tokyo (application number 29-74-A0215), and Iwate Tohoku Medical Megabank Organization, Iwate Medical University (application number HG H25-2). |

Note that full information on the approval of the study protocol must also be provided in the manuscript.

# Field-specific reporting

Please select the one below that is the best fit for your research. If you are not sure, read the appropriate sections before making your selection.

☒ Life sciences ☐ Behavioural & social sciences ☐ Ecological, evolutionary & environmental sciences

For a reference copy of the document with all sections, see nature.com/documents/nr-reporting-summary-flat.pdf

# Life sciences study design

All studies must disclose on these points even when the disclosure is negative.

| | |
|---|---|
| Sample size | The sample size of GWASs in this study is summarized in Supplementary Table 1 and 7. We did not perform sample size calculation, and the sample size is determined by the maximum number of individuals in each cohort who passed sample QC, which is expected to increase the statistical power. To further increase the sample size, we obtained publicly available summary statistics from UK Biobank and Finngen studies, and performed a cross-ancestry meta-analysis. |
| Data exclusions | All samples were selected based on sample QC criteria for each cohort. Briefly, samples with age<18, non-EAS ancestry, or low call rate were excluded for BBJ1-180K, BBJ1-12K, BBJ2-42K and TMM-50K. Additionally, samples with amyotrophic lateral sclerosis in BBJ1-12K due to its comparatively high proportion. The detailed description is summarized in the Method section. All association analysis was performed using the dataset after QC. |
| Replication | Replication for the discovery GWAS in EAS was conducted in individuals from three independent studies, namely BBJ1-12K, BBJ2-40K, and TMM-50K. We confirmed relatively high replicability (among the nine novel lead variants associated with PUD or subtypes, four were nominally associated with PUD or its subtypes in the same direction in at least two replication datasets and five novel loci were replicated in the population-based TMM dataset). We further conducted the cross-ancestry comparison of the effect sizes. |
| Randomization | We did not apply randomization in this study since this is a genotype-phenotype association study. All the samples passed the sample QC were included in the analysis. For SNV association tests performed in this study, age, sex and top 10 principal components were adjusted in |

| | the regression. |
|---|---|
| Blinding | We did not apply blinding in this study since this is a genotype-phenotype association study. No intervention was involved in this study. |

# Reporting for specific materials, systems and methods

We require information from authors about some types of materials, experimental systems and methods used in many studies. Here, indicate whether each material, system or method listed is relevant to your study. If you are not sure if a list item applies to your research, read the appropriate section before selecting a response.

## Materials & experimental systems

| n/a | Involved in the study |
|---|---|
| ☒ ☐ | Antibodies |
| ☒ ☐ | Eukaryotic cell lines |
| ☒ ☐ | Palaeontology and archaeology |
| ☒ ☐ | Animals and other organisms |
| ☒ ☐ | Clinical data |
| ☒ ☐ | Dual use research of concern |

## Methods

| n/a | Involved in the study |
|---|---|
| ☒ ☐ | ChIP-seq |
| ☒ ☐ | Flow cytometry |
| ☒ ☐ | MRI-based neuroimaging |

