## [Peer Review File · Nature Genetics]

Peer Review Information

Manuscript Title: East Asian-specific and cross-ancestry genome-wide meta-analyses provide mechanistic insights into peptic ulcer disease

Corresponding author name(s): Professor Yoichiro Kamatani

Reviewer Comments & Decisions:

Decision Letter, initial version:

17th January 2023

Dear Yoichiro,

Your Article "East Asian-specific and cross-ancestry genome-wide meta-analyses provide mechanistic insights into peptic ulcer disease" has been seen by three referees. You will see from their comments below that, while they find your work of interest, they have raised some relevant points. We are interested in the possibility of publishing your study in Nature Genetics, but we would like to consider your response to these points in the form of a revised manuscript before we make a final decision on publication.

To guide the scope of the revisions, the editors discuss the referee reports in detail within the team, including with the chief editor, with a view to identifying key priorities that should be addressed in revision, and sometimes overruling referee requests that are deemed beyond the scope of the current study. In this case, we ask that you address all technical queries related to the association analyses and their interpretation, revising the presentation for clarity where needed and extending the analyses where feasible as requested by the referees. We hope you will find this prioritized set of referee points to be useful when revising your study. Please do not hesitate to get in touch if you would like to discuss these issues further.

We therefore invite you to revise your manuscript taking into account all reviewer and editor comments. Please highlight all changes in the manuscript text file. At this stage, we will need you to upload a copy of the manuscript in MS Word .docx or similar editable format.

*2) If you have not done so already, please begin to revise your manuscript so that it conforms to our Article format instructions, available [here](http://www.nature.com/ng/authors/article_types/index.html). Refer also to any guidelines provided in this letter.

[redacted]

We hope to receive your revised manuscript within 8-12 weeks. If you cannot send it within this time, please let us know.

Nature Genetics is committed to improving transparency in authorship. As part of our efforts in this direction, we are now requesting that all authors identified as 'corresponding author' on published papers create and link their Open Researcher and Contributor Identifier (ORCID) with their account on the Manuscript Tracking System (MTS), prior to acceptance. ORCID helps the scientific community achieve unambiguous attribution of all scholarly contributions. You can create and link your ORCID from the home page of the MTS by clicking on 'Modify my Springer Nature account'. For more information, please visit www.springernature.com/orcid.

Sincerely,
Kyle

Kyle Vogan, PhD
Senior Editor
Nature Genetics
<https://orcid.org/0000-0001-9565-9665>

Referee expertise:

Referee #1: Genetics, statistical methods, cross-ancestry analyses

Referee #2: Genetics, gastrointestinal diseases

Referee #3: Genetics, gastrointestinal diseases

Reviewers' Comments:

Reviewer #1:
Remarks to the Author:

The authors conducted a large-scale GWAS of peptic ulcer disease (PUD) in the Asian ancestry and a cross-ancestry meta-analysis including the European ancestries. They found many novel loci associated with PUD and examined the cross-ancestry and cross-subtype genetic heterogeneity. The analyses were further extended to investigate the impact of *H. pylori* on PUD etiology. The authors also characterized the molecular effect of the genetic loci associated with PUD through eQTL and pQTL analyses. In general, this is an important study to understand the genetics of PUD outside of European populations. Here are my comments for the authors to consider.

1. Quality control-related comments

a) Has the sample overlap and relatedness across BBJ1-180K, BBJ1-12K, BBJ2-40K, and TMM-50K been characterized?

b) I don't think the samples were QC'ed by inbred coefficients (f_{het}), which is a typical QC measure. Why is that?

c) Another typical QC measure on the variants, the test for different missingness in cases and controls, is also missing.

2. Fine-mapping (SuSiE) analysis: What dataset was the fine-mapping analysis performed on? Was it on BBJ1-180K or the meta-analysis with all BBJ samples?

3. The analyses on GC appear ad hoc (lines 253-268). What is the purpose of this analysis? If the goal is to test whether this study supports the hypothesis that DU is protective of GC, shouldn't a Mendelian Randomization analysis be a better fit?

4. In several figures, such as SF7, a normalization (“normalized beta values”) was mentioned but not explained (e.g., whether the normalization was done for columns or rows).
5. How does the winner’s curse affect the effect size comparison across ancestries?
6. I can’t find the sample size for the HP stratified analysis. I wonder whether the HP-positive specific association can result from power rather than biology. (lines 212 -213)
7. The colocalization analysis appears confusing. For example, why only two loci were picked and undergo this analysis? I also wasn’t sure how to interpret ST19 for the results authors presented in lines 214-219. For example, while I am familiar with the coloc analysis, I don’t know how the table should be interpreted to support “the most significant loci at PSCA were independent of HP infection, marking a strong impact on the onset of PUD regardless of HP infection status”?
8. Some of the columns in the supplementary tables should have better annotations. For example, what is the LD_R2 in ST16? Was it calculated using EAS or EUR?
9. The Pi_{GU} and Pi_{DU} in line 239: is the difference of statistical significance?
10. PRS analysis: isn’t it customary to report the performance as liability threshold R2? (as in the PRS-CS method paper)
11. I’m concerned that the authors did simple lookups of the top PUD-associated variants in various datasets such as pheWAS, eQTL, and pQTL. Due to the LD, I suspect some of the disease variants showing significance in these datasets can be just tagging the causal variants for these datasets, not itself causal.
12. Table 1: it would be good to add the allele frequency to this table
13. Data sharing: it appears the summary statistics for TMM-50K will not be openly shared. The HP stratified analysis can be very helpful if shared. I’d urge the authors to open the sharing for this summary statistics.

Reviewer #2:
Remarks to the Author:

line 96 - Fixed-effect meta-analyses were used. PUD is a disease in which many factors can differ between cohorts (NSAID use, Hp infection etc.). Can the authors objectify its appropriateness in contrast to the random effects model?

Genomic coordinates were mapped against GRCh37. Many would feel that it is strongly recommend switching to GRCh38/hg38, if you are working with human sequence data. In addition to adding many alternate contigs, GRCh38 corrects thousands of small sequencing artifacts that cause false SNPs and indels to be called when using the GRCh37 assembly (b37/Hg19). It also includes synthetic centromeric sequence and updates non-nuclear genomic sequence. Are results meaningfully different if GRCh38/hg38 would be used?

Line 132 - in view of the still relatively low P value for DU across ancestries, maybe "noteworthy" is a bit too strong.

line 145 - The authors conducted a fine-mapping analysis using SuSiE to identify the causal variants. In lower power situations, even Bayesian fine-mapping methods that simultaneously model causal variants may identify a single SNP which tags two or more causal variants and the interpretation of non-colocalisation at such false signals is likely to be misleading. Can the authors say something whether this might have influenced results?

line 193 - "Although not statistically significant, GU showed a positive genetic correlation with GC, whereas DU was negatively correlated" - what were the (two-sided?) significance values observed?

line 270 - MAGMA - there has not been a universally accepted value of the sliding window size around genes and this may need several rounds of optimization. In addition, there are some recent discussions related to the statistical stability of the H-MAGMA analysis and possible fallout (ref 35). Hence the strategy taken here requires more explanation.

Discussion - In view that Hp infection is a main risk factor for PUD, how do the results relate to known risk factors for such infection (i.e. the Toll-like receptor (TLR1/6/10))? Would the difference in population be related to this? Similar points could be made for other risk factors, e.g. smoking.

- "Our single cell analysis further revealed the association of serotonin-secreting EC cells, somatostatin-secreting stomach D cells, and stomach tuft cells with PUD, indicating their key role in PUD etiology" - the exact (potential) roles and interactions of these cells in PUD remains very undefined in the text and could do with further elaboration.

Reviewer #3:
Remarks to the Author:

The authors conducted a large-scale cross-ancestry meta-analysis of genome-wide association studies of peptic ulcer disease. This was done by combining data from different population-based biobanks (Biobank Japan, Tohoku Medical Megabank, UK Biobank and FinnGen) followed by a series of downstream annotation of the identified variants based on publicly available resources. The analyses are for the most part state-of-the-art for (cross-ancestry) GWAS analyses.

Major comments:

- The majority of the samples have been included in previous published analyses which also studied peptic ulcer disease. The BBJ, FinnGen and UKBB samples have been analyzed and published as part of the large study including 220 human phenotypes published in Nature Genetics in 2021 (Sakue et al. PMID: 34594039) and the UK Biobank and FinnGen have been queried for peptic ulcer disease specifically in a study in Nature Communications in 2021 (Wu et al. PMID: 33608531). The authors should make very clear which samples are new for the current analyses.

- The case definition in the TMM-50K is based on self-reported PUD. While individuals might potentially

reliably report peptic ulcer disease, the distinction between gastric or duodenal ulcers will be very unreliable if based on questionnaires.

- The major causes for peptic ulcer disease are *Helicobacter pylori* infection and the use of NSAIDs. The authors do perform a *H. pylori* stratified analyses, but this is only done in the TMM-50K cohort (based on self-reported PUD) and not in the other cohorts. The diagnosis of PUD in the BBJ cohorts is made based on interviews and reviewing medical records. Is there more detail on the cause of peptic ulcer disease available, e.g. HP or NSAID?

- Overall, they identify 25 novel loci that are concordant across ancestries. But the manuscript is by times difficult to read, listing numbers of associated genetic variants for many different analyses, e.g. in PUD, GU, DU, BU either in East Asian populations or in combined populations. It would be helpful to present these results better. Maybe in a figure that makes clear which result come from which analyses and which results are novel.

- The paper is heavily focused on the computational analyses and downstream annotation. It lacks a good discussion on the biological interpretation of the identified variants and potential implications.

Author Rebuttal to Initial comments

Dear Dr. Kyle Vogan,

We thank you very much for inviting us to revise our manuscript, NG-A61277: "East Asian-specific and cross-ancestry genome-wide meta-analyses provide mechanistic insights into peptic ulcer disease."

We appreciate the insightful and comprehensive comments from the reviewers. Based on these highly constructive comments, we have substantially revised our manuscript. We believe the manuscript is considerably improved after addressing all remarks of the reviewers.

Major updates in the revised manuscript:

1. In order to address the technical queries related to association studies, we have applied additional QC procedures, extended current analyses, and clarified the ambiguous points, which strengthened our findings and improved the interpretations. We have tried to address the reviewers' concerns about the effect of potential sample overlap, sample heterozygosity rate, and non-random missingness of SNP calling on our results; we showed that the potential impacts of these factors on our results were negligible. Further, we performed the winner's curse correction and power analysis to support the findings from per-allele effect size comparisons in cross-ancestry meta-analysis and HP-stratified analysis in TMM-50K.
2. We performed additional analyses to support the robustness of the methods used in this study. First, we performed a meta-analysis of PUD in EAS using the random-effect model; compared with the analysis using the fixed-effect model, we observed concordant results from the random-effects model in terms of significant loci (15 out of 17 loci). Second, we

conducted additional imputation using a GRCh38-based reference panel (1000 Genomes Project 30X high coverage dataset) for BBJ1-12K and BBJ2-42K, and performed cross-ancestry meta-analysis using the GRCh38-based dataset. The GRCh38-based meta-analysis showed high consistency with the GRCh37-based study for shared SNPs, and no additional GRCh38-specific loci were identified. Third, we performed further MAGMA analysis with a range of window sizes around the gene for gene-based analysis, tissue-specificity, and cell-specificity analysis. The results utilizing different window sizes showed relatively high correlations for all analyses, and the selection of window sizes only marginally affected the results.

3. To improve the interpretation of the novel findings in this study, we restructured the discussion section and added detailed biological interpretations of the mechanisms of identified variants and the potential roles of cell types on PUD etiology, especially for D cells and EC cells. We discussed how PUD genetically correlated with its risk factors (i.e., smoking) and other diseases (immune-related diseases). To clarify the source of datasets and identified associations, we added supplementary materials to address the potential ambiguity.
4. We advanced the process to release the summary statistics for overall and *H.pylori*-stratified analysis in TMM-50K, which we expected to be helpful for researchers in a related field.

Additionally, we revised the legends and annotations of several figures and tables in this manuscript to improve the general readability and interpretability, as the reviewers suggested.

Please find the point-by-point responses to all comments from the reviewers below. (Responses are surrounded by black borders, and revised contents in the manuscript are highlighted in blue.)

All co-authors have read the revised manuscript and approved this submission.

We hope this revised manuscript is suitable for publication in Nature Genetics.

Thank you very much for your consideration.

Best regards,

Point-to-point response

Referee expertise:

Referee #1: Genetics, statistical methods, cross-ancestry analyses

Referee #2: Genetics, gastrointestinal diseases

Referee #3: Genetics, gastrointestinal diseases

Reviewers' Comments:

Reviewer #1:

Remarks to the Author:

The authors conducted a large-scale GWAS of peptic ulcer disease (PUD) in the Asian ancestry and a cross-ancestry meta-analysis including the European ancestries. They found many novel loci associated with PUD and examined the cross-ancestry and cross-subtype genetic heterogeneity. The analyses were further extended to investigate the impact of *H. pylori* on PUD etiology. The authors also characterized the molecular effect of the genetic loci associated with PUD through eQTL and pQTL analyses. In general, this is an important study to understand the genetics of PUD outside of European populations. Here are my comments for the authors to consider.

We thank Reviewer #1 for the very precise and accurate summary of our work and the insightful and constructive comments on this study.

1. Quality control-related comments

We thank the reviewer for the comments and suggestions on this study's quality control (QC) procedures. The genotype QC methods in this study are in alignment with previous genome-wide association studies conducted in Biobank Japan (Sakaue, S. et al. Nature Genet. 2021, PMID 34594039; Ishigaki, K. et al. Nat Genet. 2020, PMID 32514122; Akiyama, M. et al. Nat Commun. 2019, PMID 31562340).

We rephrased the description to clarify the QC methods used in this study:

(Methods, line 564)

QC of autosomal genotypes was performed as described previously⁵⁸. Briefly, we ...

a) Has the sample overlap and relatedness across BBJ1-180K, BBJ1-12K, BBJ2-40K, and TMM-50K been characterized?

Biobank Japan Project first cohort (BBJ1) and second cohort (BBJ2) both recruited unique individuals from participating hospitals across Japan. For BBJ2, new individuals were recruited to avoid sample overlap with BBJ1 at each participating hospital (<https://biobankjp.org/en/index.html>). The Population-based cohort in Tohoku Medical Megabank Project (TMM) recruited only residents in Iwate and Miyagi prefectures (<https://www.megabank.tohoku.ac.jp/english/research/cohortbiobank/>). Due to ethical reasons, we cannot directly estimate the sample overlap between BBJ and TMM.

However, we cannot rule out the possibilities that the same individuals might visit two or more participating hospitals or that the cohorts might include each of monozygotic twins. To confirm the sample overlap within BBJ1 and BBJ2, of which we have access to genotypes, we merged the datasets of both cohorts and checked the relatedness among individuals in BBJ1-180K, BBJ1-12K, and BBJ2-42K using PLINK (PI_HAT>0.75 as the threshold for potentially overlapping samples). We observed a total of 137 pairs across BBJ1-180K, BBJ1-12K, and BBJ2-42K, which only account for 0.057% (137 / 236,331) of the total samples. Thus, the sample overlap in this study is limited. Additionally, LD score regressions suggested no substantial inflation for EAS-specific meta-analysis analysis (intercepts ranging from 1.01 to 1.02 for PUD and its subtypes; **Supplementary Table 5**). Based on these results, we consider the influence of sample overlap for our meta-analysis negligible in such situations.

We modified the descriptions in the methods accordingly. Additionally, we corrected the inconsistent denotation of BBJ2-40K and BBJ2-42K to BBJ2-42K.

(Methods, lines 535 - 537)

Sample overlap was checked in BBJ1-180K, BBJ1-12K, and BBJ2-42K by merging three datasets and estimating the identical-by-descent (IBD) sharing, and the number of potentially overlapping samples is minimal (< 0.06%; **Supplementary Figure 32**).

(Results, lines 86 - 88)

Replication was conducted in individuals from three independent studies, namely BBJ1-12K (1,001 cases), BBJ2-42K (3,637 cases), and TMM¹³-50K (a population-based study; 5,388 cases; **Supplementary Table 1; Methods**).

(Supplementary figures)

Supplementary Figure 32. Venn Plot of the potential sample overlap within Biobank Japan cohorts.

Sample overlap was estimated by IBD sharing.

b) I don't think the samples were QC'ed by inbred coefficients (f het), which is a typical QC measure. Why is that?

We thank the reviewer for this comment and apologize for not clearly describing the sample QC procedures. The distributions of F coefficients for the samples used in this study are shown in Response Figure 1. For BBJ1-12K, and BBJ2-42K, we had processed the datasets independently and already removed heterozygosity outliers with +/- 4 standard deviations (SD) from the mean F, which is a commonly used threshold in biobank-scale cohorts as in Kurki, M. I. et al. Nature. 2023, PMID 36653562. For BBJ1-180K, we used the same QC'ed genotype dataset described in previous papers (Sakaue, S. et al. Nature Genet. 2021, PMID 34594039; Ishigaki, K. et al. Nat Genet. 2020, PMID 32514122; Akiyama, M. et al. Nat Commun. 2019, PMID 31562340). We confirmed that no individuals in BBJ1-180K have an F coefficient less than minus four standard deviations (SD) from the mean (which indicates a higher heterozygosity rate). These results suggested no substantial DNA contamination in this study.

Due to the large sample size, a small proportion of individuals in BBJ1-180K (1.4%) have an F coefficient higher than mean+4SD, which reflected the fine-scale population structures in certain regions of Japan (Sakaue, S. et al. Nat Commun. 2020, PMID 32218440). We did not exclude samples based on the high F coefficient as in previous studies conducted in BBJ. Generalized linear mixed model-based methods were employed to maximize the sample size, as Sakaue, S. et al. described (Nature Genet. 2021, PMID 34594039). LD score regressions also suggested that the statistics were not substantially inflated by population stratification or cryptic relatedness (λ_{GC} ranging from 1.014 to 1.059 and intercept ranging from 0.996 to 1.006 for PUD and its subtypes in BBJ1-180K; **Supplementary Table 5**).

We revised the methods to clarify the QC procedures.

(Response figures)

Response Figure 1. Distribution of F coefficient in Biobank Japan.

a, Distributions of F coefficient in BBJ1-180K, BBJ1-12K and BBJ2-42K. Blue lines show the mean F coefficient in each cohort. Dashed lines represent the mean \pm 4SD. b, Scatterplot of PC1 and PC2 for individuals in BBJ1-180K. Markers are colored by the F coefficient.

(Methods lines 563 - 573).

We confirmed no sample in BBJ1-180K had excess heterozygosity (4 standard deviations (SD) from the mean). QC of autosomal genotypes was performed as described previously⁶². Briefly, we excluded the genotyped variants based on the following criteria for BBJ1-180K: (1) call rate $< 99\%$, (2) heterozygote count < 5 , (3) Hardy–Weinberg-equilibrium $P < 1.0 \times 10^{-6}$, and (4) concordance rate $< 99.5\%$ or non-reference discordance rate $\geq 0.5\%$ between array genotypes and whole-genome-sequence dataset using overlapping participants ($n = 939$), as described previously⁶². We applied the same sample and variant QC criteria (except (4) for autosomal genotype QC) as in the discovery stage to the replication sets of BBJ1-12K and BBJ2-42K. Additionally, we removed samples with extreme heterozygosity rate (± 4 SD from the mean) in BBJ1-12K and BBJ2-42K, and the samples with amyotrophic lateral sclerosis in BBJ1-12K due to its comparatively high proportion.

c) Another typical QC measure on the variants, the test for different missingness in cases and controls, is also missing.

We thank the reviewer for the comment. We think that the test for non-random missingness is commonly employed when cases and controls are collected separately. (Lee, S. H. et al. Am J Hum Genet. 2011, PMID 21376301) For both BBJ and TMM, all samples were collected using the same procedure without considering the case/control status for PUD.

We additionally performed tests for different missingness in cases and controls for BBJ1-180K, BBJ1-12K, and BBJ2-42K using the datasets genotyped by SNP arrays. After multi-testing correction, only one variant in BBJ1-180K showed a significant difference (rs4258597, $P < 0.05/520135$).

Based on the results above and considering that we applied a relatively strict genotype missing threshold (excluding variants with missing rate > 0.01) for BBJ datasets, we believe that the effect of the significant difference in missingness between cases and controls for the variant is negligible in this study.

We modified the texts and added a supplementary table accordingly.

(Supplementary tables)

Supplementary Table 33. Variants with significant non-random missingness in BBJ cohorts.

Cohort	CHR	rsID	Missing rate in cases	Missing rate in controls	Fisher's exact test P
BBJ1-180K	16	rs4258597	0.00137	0.000222	1.04E-10

(Methods, lines 573 - 574)

Non-random missingness in cases and controls was tested in BBJ1-180K, BBJ1-12K, and BBJ2-42K (**Supplementary Table 33**).

2. Fine-mapping (SuSiE) analysis: What dataset was the fine-mapping analysis performed on? Was it on BBJ1-180K or the meta-analysis with all BBJ samples?

We apologize for the lack of clarity in the original manuscript. We used the summary statistics of meta-analysis in EAS (including BBJ1-180K, BBJ1-12K, BBJ2-42K, and TMM-50K) to increase the statistical power. The in-sample LD was calculated using unrelated individuals (KING kingship coefficient value < 0.0884 , $N = 171,085$) from BBJ1-180K.

We modified the manuscript to clarify this.

(Methods, lines 710 – 712)

Unrelated individuals (KING kingship coefficient value < 0.0884 , $N = 171,085$) from BBJ1-180K were used as LD reference, computed by LDstore (v2.0; <http://www.christianbenner.com/>)⁷¹ based on the imputed dosages.

3. The analyses on GC appear ad hoc (lines 253-268). What is the purpose of this analysis? If the goal is to test whether this study supports the hypothesis that DU is protective of GC, shouldn't a Mendelian Randomization analysis be a better fit?

We thank the reviewer for the suggestions and apologize for the confusion on the analysis of gastric cancer (GC). The analysis aims to investigate the epidemiological findings of the opposite associations of GU and DU with GC from a genetic perspective. We agree with the reviewer on this point. To address this issue, we conducted a two-sample Mendelian randomization (MR) to investigate the causal relationship between PUD (including subtypes) and GC.

To avoid sample overlap between the exposure and outcome, we conducted additional meta-analysis combining GWASs for PUD and its subtypes in BBJ1-12K, BBJ2-42K, and TMM-50K, leading to 23 available independent variants (instrumental variables). To avoid bias caused by weak instruments, we estimated per-variant F statistics for each exposure and removed variants with $F < 10$ from the instrumental variables for the exposure. The summary statistics for GC were obtained from the previous study conducted in BBJ1-180K (Ishigaki, K. et al. Nat Genet.

2020, PMID 32514122). The potential sample overlap between BBJ1-180K and BBJ1-12K+BBJ2-42K was extremely limited (related to comment 1a).

The MR analysis was performed using TwoSampleMR (Hemani, G. et al. eLife. 2018, PMID 29846171) to evaluate the causality of PUD or its subtypes on GC. Although PUD and its subtypes showed significant ($p < 0.05/12$) protective effects against GC in the analysis using the Inverse variance weighted (IVW) method, MR-Egger analysis suggested significant pleiotropy for the instruments. We applied MR-PRESSO (Verbanck, M. et al. Nat Genet. 2018, PMID 29686387) to detect and correct for the horizontal pleiotropic variants (ranging from 6 to 7 for each exposure). After the corrections, we observed no significant effects of PUD or its subtypes on GC.

These results suggest that several variants (identified by MR-PRESSO, listed in **Supplementary Table 26**) might directly have horizontal pleiotropic effects on both PUD (including subtypes) and GC. Further research that increases the statistical power of MR is warranted to prove the genetic evidence from epidemiological studies that DU is likely to be protective against GC.

We modified the texts in the manuscript and added supplementary tables accordingly. These updates are reflected in the following:

(Results, lines 253 - 268)

Pleiotropy of PUD risk variants on GC

Considering that DU appears to be a protective factor against GC, we conducted two-sample Mendelian randomization (MR)³⁴ to evaluate the causality of PUD or its subtypes on GC. Summary statistics for GC in EAS were obtained from a previous study in BBJ1-180K, which included approximately 6,500 cases³⁵ (Methods). Summary statistics for PUD and its subtypes were obtained by conducting a meta-analysis combining three replication datasets. Although PUD and its subtypes showed significant ($P < 0.05/15$) protective effects against GC using the Inverse variance weighted (IVW) method (**Supplementary Table 24**), MR-Egger analysis suggested significant pleiotropy for the instruments (**Supplementary Table 25**). MR-PRESSO³⁶ was utilized to correct for the horizontal pleiotropic variants (ranging from 6 to 7 for each exposure). The outlier-corrected MR showed no significant effects of PUD or its subtypes on GC

(Supplementary Table 26). We note that splitting samples and removing outliers may cause power loss (Supplementary Figure 24).

To evaluate the pleiotropic effects of PUD risk variants on GC risk, we compared the effect sizes of distinct signals in EAS between PUD subtypes and GC.

(Discussion, lines 342 - 353)

We found three variants (linked to *EFNA1*, *PTGER4*, and *PSCA*) to have relatively large pleiotropic effects on DU and GC. ... No causal effects were identified in the outlier-corrected MR in this study, which may suffer from the power loss caused by sample split and removal of variants. Further research is warranted on the protective role of DU against GC.

(Methods, lines 779 - 791)

Two-sample Mendelian randomization analysis

MR analysis was performed using TwoSampleMR³⁴ to evaluate the causality of PUD or its subtypes on GC. To avoid sample overlap between the exposure and outcome, we conducted additional meta-analysis combining GWASs for PUD and its subtypes in BBJ1-12K, BBJ2-42K, and TMM-50K. A total of 23 available independent variants identified in the EAS-specific meta-analysis were used as instrumental variables. To avoid bias caused by weak instruments, we further estimated per-variant F statistics for each exposure and removed variants with $F < 10$ from the instrumental variables for the exposure. MR-PRESSO³⁶ was employed to correct for the horizontal pleiotropic variants. The summary statistics for GC were obtained from the previous study conducted in BBJ1-180K³⁵. Statistical power for MR was approximately estimated using methods described previously⁷⁸. Compared with MR using all available variants, the pleiotropic outlier-correction may result in insufficient power (Supplementary Figure 24).

(Supplementary tables)

Supplementary Table 24. Two-sample Mendelian randomization study of the causality of PUD or its subtypes on GC.

Supplementary Table 25. Tests for directional pleiotropy and heterogeneity in MR.

Supplementary Table 26. Outlier-corrected MR using the MR-PRESSO method.

(Supplementary figures)

Supplementary Figure 24. Statistical power estimation for Mendelian randomization analysis. The proportion of variance (r^2) explained by genetic instruments was approximated by the sum of the explained variance by the selected variants. Grey dashed lines represent the sample sizes. Significance, 0.0166; odds ratio, 1.2. a, Power estimation for the analysis of PUD; the ratio of cases to controls, 1:8.7; sample size, 97,523. b, Power estimation for the analysis of DU; the ratio of cases to controls, 1:18; sample size, 92,294. c, Power estimation for the analysis of GU; the ratio of cases to controls, 1:13; sample size, 94,056.

4. In several figures, such as SF7, a normalization (“normalized beta values”) was mentioned but not explained (e.g., whether the normalization was done for columns or rows).

We apologize for the insufficient explanation of the figures. The normalized beta values in SF7 were obtained from the GTEx project (GTEx Consortium. Science. 2020, PMID 32913098), and no further normalization was performed in this study. The normalized beta refers to the linear regression slope implemented in the FastQTL software (Ongen, H. et al. Bioinformatics. 2015, PMID 26708335), which is employed in the GTEx project for eQTL analysis. Expression levels were normalized before the regression (<https://www.gtexportal.org/home/faq#interpretEffectSize>).

We have revised the legends and added explanations accordingly.

(Supplementary figures)

Supplementary Figure 9. (Partially extracted)

Square colors represent the normalized beta values (slope of the linear regression in eQTL mapping) of the eQTL allele that is in LD with the PUD risk allele.

5. How does the winner's curse affect the effect size comparison across ancestries?

We thank the reviewer for this comment. We compared the winner's curse (WC)-corrected effect sizes of the lead variants identified in the EAS meta-analysis in the current study and the variants reported in UKB⁹ using the methods described previously (Zhong, H. et al. Biostatistics. 2008, PMID 18310059; Palmer, C. et al. PLoS Genet. 2017, PMID 28715421). This method adjusts effect size estimators by maximizing the conditional likelihood at observed effect sizes.

As shown in **Supplementary Figure 7**, compared with the uncorrected correlation ($r=0.80$), the trend remains ($r=0.77$) after the WC correction, which suggested the effect of WC was relatively limited ($\Delta r = 0.03$) in this cross-ancestry comparison.

We added texts and a supplementary figure accordingly.

(Supplementary figures)

Supplementary Figure 7. Effect size comparison of lead variants with and without the winner's curse corrections.

a, comparison using lead variants of significant loci ascertained in EAS. b, the winner's curse-corrected comparison using lead variants of significant loci ascertained in EAS. c, comparison using lead variants of significant loci ascertained in EUR. d, the winner's curse-corrected comparison using lead variants of significant loci ascertained in EUR.

(Results, lines 125 - 126)

The correlation remained after the winner's curse corrections. (Methods; Supplementary Figure 7)

(Methods, lines 816 - 818)

Additionally, we compared the winner's curse (WC)-corrected effect sizes of the lead variants identified in the EAS meta-analysis in the current study and the variants reported in UKB⁵ using the methods as described previously^{80,81}.

6. I can't find the sample size for the HP stratified analysis. I wonder whether the HP-positive specific association can result from power rather than biology. (lines 212 -213)

We apologize for the confusion. The sample size for HP-stratified analysis is listed in **Supplementary Table 19**. Additionally, we estimated the statistical power with the effect size observed in HP-positive PUD GWAS for the variant (*CCKBR*) that showed significant heterogeneity using the Genetic Association Study (GAS) Power Calculator (https://csg.sph.umich.edu/abecasis/gas_power_calculator/; Skol, A. D. et al. Nat Genet. 2006, PMID 16415888).

With such settings (Controls, 26,432; Significance level, 0.05/31; Disease model, Additive; Prevalence, 0.109; Disease Allele Frequency, 0.093; Genotype Relative Risk, 1.189.

), the statistical power was estimated to be 0.881 for HP-negative PUD GWAS (Ncase, HP-negative = 3,372). Based on the results showing enough statistical power (Power > 0.8), the HP-positive specific association is more likely to result from biological reasons than the power issue for HP-negative PUD GWAS.

(Methods, lines 733 - 735)

Since stratified analysis could reduce power and lead to false negative results, we estimated the power of GWAS of HP-negative PUD for the identified HP-positive-specific variant. (**Supplementary Figure 36**).

(Supplementary figures)

Supplementary Figure 36. Power analysis for GWAS of HP-negative PUD in TMM-50K.

Statistical power was estimated using GAS Power Calculator (https://csg.sph.umich.edu/abecasis/gas_power_calculator/). The dashed line represents the number of cases ($N_{\text{case, HP-negative}} = 3,372$) used for GWAS of PUD in HP-negative individuals from TMM-50K. Other settings: the number of controls, 26,432; significance level, 0.0016; disease model, additive; prevalence, 0.109; disease allele frequency, 0.093; genotype relative risk, 1.189.

7. The colocalization analysis appears confusing. For example, why only two loci were picked and undergo this analysis? I also wasn't sure how to interpret ST19 for the results authors presented in lines 214-219. For example, while I am familiar with the coloc analysis, I don't know how the table should be interpreted to support "the most significant loci at PSCA were independent of HP infection, marking a strong impact on the onset of PUD regardless of HP infection status"?

We apologize for the confusion. Our initial workflow for this analysis was (1) we compared the effect sizes of all lead variants identified in the EAS-specific meta-analysis for GWAS in HP-positive individuals and in HP-negative individuals to identify the variants showing significant

heterogeneity. (2) we next conducted colocalization analysis for the HP-stratified GWASs to test the HP-negative PUD and HP-positive PUD shared the same causal variants.

In the effect size comparison, we did not identify a significant difference in the effect size of the lead variants in the *PSCA* locus. For the *PSCA* locus, as shown in the updated **Supplementary Table 20**, these results suggested HP-negative PUD and HP-positive PUD shared the same causal variant (PP.H4: 0.72 ~ 0.95; PP.H3+PP.H4>0.8 for PUD and all subtypes), indicating that the signal is likely to be independent of HP infection. We additionally applied coloc to the signals identified by SuSiE for this locus. SuSiE identified one signal for HP-negative PUD and one for HP-positive PUD; the two signals were colocalized (PP4>0.8).

Despite enough power to detect the identified variant in association tests (related to comment #6), the lack of signals identified by SuSiE and the low PP3 and PP4 for other loci suggested insufficient power for colocalization analysis.

To clarify this, we revised the manuscript and added results for all loci in Supplementary Table 20 accordingly.

(Results, lines 215 - 218)

On the other hand, the lead variants in the most significant locus at *PSCA* did not show significant heterogeneity in effect sizes ($P < 0.05$). Colocalization analysis suggested HP-positive PUD and HP-negative PUD shared the causal variant in *PSCA* (PP4>0.8 for PUD; **Supplementary Table 20**).

(Methods, lines 739 - 740)

For loci with multiple independent signals identified in the conditional analysis, coloc was applied to the signals identified by SuSiE.

(Discussion, line 326)

Our results also showed the signal at *CCKBR* (receptor for gastrin) to be HP-positive-specific.

(Discussion, lines 366 - 368)

Despite the biobank-scale cohort for HP-stratified analysis, the statistical power is still limited for certain analyses.

(Supplementary tables)

Supplementary Table 20 (partially extracted). Colocalization analysis for HP-stratified analysis.

Analysis	Phenotype	NSNPS	Locus	PP.H0.abf	PP.H1.abf	PP.H2.abf	PP.H3.abf	PP.H4.abf	Signal for trait 1	Signal for trait 2
coloc	PUD	5695	8:143761931:C:T	0.0000	0.0000	0.0000	0.0448	0.9552	/	/
coloc	GU	5685	8:143761931:C:T	0.0000	0.0716	0.0000	0.2029	0.7255	/	/
coloc	DU	5686	8:143761931:C:T	0.0000	0.0000	0.0000	0.0380	0.9620	/	/
coloc	BU	5674	8:143761931:C:T	0.0000	0.0299	0.0000	0.0827	0.8874	/	/
coloc-susie	PUD	11729	8:143761931:C:T	0.0000	0.0000	0.0000	0.0874	0.9126	8:143761931:C:T	8:143771714:C:A
coloc-susie	DU	11723	8:143761931:C:T	0.0000	0.0000	0.0000	0.0752	0.9248	8:143761931:C:T	8:143771712:A:C

8. Some of the columns in the supplementary tables should have better annotations. For example, what is the LD_R2 in ST16? Was it calculated using EAS or EUR?

We thank the reviewer for the comments. We revised the descriptions in the supplementary tables accordingly. For LD_R2 in ST16, LD_R2 shows the LD R2 between the lead variants and pQTLs in the populations indicated in the Population column (the highest LD R2 in EAS or EUR used for tagging).

The updates can be found in **Supplementary Table 16**.

(Supplementary tables)

Supplementary Table 16. Overlap of lead variants associated with PUD and PUD subtypes in EAS with pQTL signals.

9. The Pi_GU and Pi_DU in line 239: is the difference of statistical significance?

We thank the reviewer for this question. Using the Bayesian mixed linear model implemented in SBayesS, we estimated the posterior means of P_i (Polygenicity; the proportion of SNP with non-zero effect) for GU and DU. We also note that these polygenicity estimates are relative and cannot be interpreted as the number of causal variants (Zeng, J. et al. Nat Commun. 2021, PMID 33608517). Since the same set of variants was used in SBayesS for all phenotypes, we used these estimates as a relative quantification of the degree of polygenicity. Given that $P_{i_{DU}}$ and $P_{i_{GU}}$ are both Bayesian estimators, we compared the posterior distributions of $P_{i_{DU}}$ and $P_{i_{GU}}$. As shown in **Supplementary Figure 23**, the 90% credible intervals of $P_{i_{DU}}$ (0.00069, 0.00154) and $P_{i_{GU}}$ (0.00151, 0.00339) only marginally overlapped, which indicated a low probability that true values of $P_{i_{DU}}$ and $P_{i_{GU}}$ were within the same range. This suggested that $P_{i_{DU}}$ and $P_{i_{GU}}$ were most likely to be different, with $P_{i_{GU}}$ being larger than $P_{i_{DU}}$.

We toned down the difference in polygenicity and revised the manuscript to clarify this.

(Results, lines 237 - 239)

Compared to DU, GU showed **moderately** higher polygenicity but lower heritability ($P_{i_{GU}} = 0.24\%$, $P_{i_{DU}} = 0.10\%$; **Supplementary Figure 22 - 23**; **Supplementary Table 22**).

(Methods, lines 746 - 747)

SBayesS employs a Bayesian mixed linear model and reports **the posterior means of** SNP-based heritability, polygenicity estimates, and a metric that indicates negative selection.

(Supplementary figures)

Supplementary Figure 23. Posterior distribution of polygenicity estimates for GU and DU in EAS using SbayesS.

The histogram shows the posterior distributions of the polygenicity for GU (Pi_{GU}) and DU (Pi_{DU}) estimated by SbayesS. Blue lines, the upper and lower bounds for the 90% credible interval for Pi_{DU} . Yellow lines, the upper and lower bounds for the 90% credible interval for Pi_{GU} .

10. PRS analysis: isn't it customary to report the performance as liability threshold R^2 ? (as in the PRS-CS method paper)

We thank the reviewer for the suggestion. We added the liability-scale R^2 (Lee, S. H et al. Genet Epidemiol. 2012, PMID 22714935) in **Supplementary Table 22** and revised the manuscript accordingly.

(Results, lines 247 – 250)

The strongest association was between DU PRS and DU (OR = 1.229, confidence interval, CI = 1.20–1.25, $\Delta R^2 = 0.94\%$). DU PRS ((OR = 1.08, CI = 1.06–1.10, $\Delta R^2 = 0.13\%$) showed a stronger association with GU than GU PRS (OR = 1.04, CI = 1.03–1.06, $\Delta R^2 = 0.05\%$; **Supplementary Table 22**).

(Methods, lines 777 - 778)

We evaluated the predictive ability of each PRS model by its improvement of AUC and R^2 on the liability scale⁷⁷ over a base model that includes age, sex, and top five PCs.

11. I'm concerned that the authors did simple lookups of the top PUD-associated variants in various datasets such as pheWAS, eQTL, and pQTL. Due to the LD, I suspect some of the disease variants showing significance in these datasets can be just tagging the causal variants for these datasets, not itself causal.

We thank the reviewer for the comments on the methodology employed in this study. We agree with the reviewer that the associations identified by simple lookups using LD might result from tagging the causal variants, referring to the previous paper (Giambartolomei, C. et al. PLoS Genet. 2014, PMID 24830394). Ideal methods to address this issue would be statistical methods incorporating LD information or colocalization analysis. However, due to a lack of available summary statistics for large-scale eQTL and pQTL analyses for many types of tissues in East Asians (the main population in this study, as mentioned above), colocalization studies do not apply to this study. This is one of the common limitations of GWAS in under-represented populations such as East Asians. The lookup approach is then a compromise in such situations. Despite the limitations, lookup approaches for PheWAS and e/pQTLs were successfully shown to suggest potential associations in multiple studies (Suzuki, K. et al. Nature Genet. 2019, PMID 30718926; Hautakangas, H. et al. Nat Genet. 2022, PMID 35115687; Vujkovic, M. et al. Nat Genet. 2022, PMID 35654975).

We modified the texts to clarify the potential issue of simple lookups.

(Discussion, lines 356 - 366)

Although we identified multiple associations, the current study has several potential limitations.

... Fourth, the variants identified by fine-mapping and the overlaps in association signals identified by lookup approaches might result from tagging distinct causal variants, which should be interpreted cautiously.

12. Table 1: it would be good to add the allele frequency to this table

We thank the reviewer for the suggestion and revised the table accordingly.

The updates can be found in **Table 1**.

13. Data sharing: it appears the summary statistics for TMM-50K will not be openly shared. The HP stratified analysis can be very helpful if shared. I'd urge the authors to open the sharing for this summary statistics.

We thank the reviewer for the comment on data sharing. For the TMM-50K-derived summary statistics (GWAS of PUD and its subtypes; HP-stratified analysis), the data is expected to be available at the Japanese Multi Omics Reference Panel (jMorp; <https://jmorp.megabank.tohoku.ac.jp>), which is a web-based database for the TMM project. We will share the data as soon as the necessary process for data release is finished according to the release policy of the TMM project.

We revised the data availability statement accordingly.

(Data availability, lines 923 - 925)

Summary statistics derived from TMM-50K (GWAS of PUD and PUD subtypes; HP-stratified analysis) will be available at the Japanese Multi Omics Reference Panel (jMorp, <https://jmorp.megabank.tohoku.ac.jp>; ID: TGA000011).

Reviewer #2:

We thank Reviewer #2 for the constructive comments and insightful questions on this study.

Remarks to the Author:

line 96 - Fixed-effect meta-analyses were used. PUD is a disease in which many factors can differ between cohorts (NSAID use, Hp infection etc.). Can the authors objectify its appropriateness in contrast to the random effects model?

We thank the reviewer for the question. For EAS-specific meta-analysis, we combined the GWASs of four cohorts consisting of mainly Japanese ancestry. Despite the lack of detailed information on the cause of PUD, these cohorts showed largely consistent demographic characteristics and PUD prevalence rates, as shown in **Supplementary Table 1**. Thus, we conducted a population-specific meta-analysis using the fixed-effect method, which has been employed in previous studies under similar study designs (Hautakangas, H. et al. Nat Genet. 2022, PMID 35115687; Mishra, A. et al. Nature. 2022, PMID 36180795; Spracklen, C. N. et al. Nature. 2020, PMID 32499647). As the Cochran's Q and I^2 heterogeneity tests showed (**Supplementary Table 6**, column P, Q), none of the lead variants in novel loci for PUD and its subtypes showed significant heterogeneity. In contrast, only three associations in previously known loci (out of a total of 48 associations) showed significant heterogeneity in effect size ($P_{\text{het}} < 0.05$, $I^2 > 0.75$). Further, to evaluate the heterogeneity between BBJ and TMM, we performed genetic correlation analysis for PUD using summary statistics from BBJ1-180K and TMM-50K, and the high genetic correlation between the two cohorts ($r_g = 0.89$, $p < 0.05$) suggested a low level of heterogeneity. Based on these results, we believe the fixed effect model is appropriate for this study.

(Supplementary figures)

Supplementary Figure 33. Manhattan plots for EAS-specific meta-analysis of PUD.

For variants above the top light grey dashed line ($-\log_{10}(P) > 20$), values are rescaled. Lead variants are annotated with the nearest gene name. Variants are plotted against GRCh37 (hg19). The bottom dark grey dashed line indicates the genome-wide significance threshold ($P < 5.0 \times 10^{-8}$). Variants with $-\log_{10}(P) < 2$ are omitted. a, Manhattan plot and Q-Q plot for the EAS-specific fixed-effect meta-analysis of PUD using METAL. b, Manhattan plot and Q-Q plot for the EAS-specific random-effects meta-analysis of PUD using GWAMA.

(Methods, 630 - 632)

Additionally, the random-effect model implemented in GWAMA was employed to evaluate the effect of heterogeneity (**Supplementary Note; Supplementary Figure 33**).

(Supplementary note)

Comparison of fixed-effect and random-effects models in the meta-analysis of PUD in EAS

We conducted a random-effects meta-analysis using GWAMA²³ for PUD in EAS. 15 out of the 17 significant loci from the fixed-effect meta-analysis still showed significant association ($p < 5 \times 10^{-8}$) under the random-effects model, with the other two loci being observed at the suggestive level ($p < 5 \times 10^{-6}$). We note that the analysis under the random-effects model was substantially underpowered ($\lambda_{GC} = 0.889$) compared with the fixed-effect model ($\lambda_{GC} = 1.052$). In the fixed-effect model, LD score regressions also supported no substantial inflation for the statistics obtained by the fixed-effect model ($\text{intercept}_{EAS, PUD} = 1.02$). Based on these results, we selected the fixed-effect approach for population-specific meta-analysis of PUD in this study.

Genomic coordinates were mapped against GRCh37. Many would feel that it is strongly recommend switching to GRCh38/hg38, if you are working with human sequence data. In addition to adding many alternate contigs, GRCh38 corrects thousands of small sequencing artifacts that cause false SNPs and indels to be called when using the GRCh37 assembly (b37/Hg19). It also includes synthetic centromeric sequence and updates non-nuclear genomic sequence. Are results meaningfully different if GRCh38/hg38 would be used?

We thank the reviewer for the recommendation and the question about the genome build version. Since two of the largest datasets in this study (BBJ1-180K and UKB) were genotyped with arrays and imputed on reference panels based on GRCh37 genomic coordinates, we conducted a meta-analysis based on GRCh38 combining BBJ1-12K, BBJ2-42K, and FinnGen, to confirm if there are any benefits to our study if we use the imputation panel based on GRCh38/hg38.

We processed and extracted the same 2504 samples from the 1000 Genome High Coverage datasets (GRCh38/hg38; Byrska-Bishop, M. et al. Cell. 2022, PMID 36055201) as in the 1000

Genome Phase3v5 panel (GRCh37/hg19; Auton, A. et al. Nature. 2015, PMID 26432245). BBJ1-12K and BBJ2-42K were additionally imputed using the 1000 Genome panel (GRCh38/hg38). We then conducted GWAS with the same settings and additional meta-analyses using the GRCh38-based and GRCh37-based datasets from BBJ1-12K, BBJ2-42K, and FinnGen.

As shown in the figures below, the meta-analysis using GRCh38-based datasets showed overall high consistency of $-\log_{10}(P)$ values ($r=0.92$) with GRCh37-based results and consistent genomic inflation factors λ_{GC} , with substantially high consistency for significant variants ($r=0.9991$). The GRCh38-based meta-analysis identified the same significant loci with similar significance. No additional GRCh38-specific loci were identified.

Based on these additional analyses, we believe the differences between analyses based on different reference genome builds were limited in this study.

(Supplementary figures)

Supplementary Figure 34. Manhattan plots and Q-Q plots for cross-ancestry meta-analysis.

GWASs of PUD, DU, and GU conducted in BBJ1-12K, BBJ2-42K, and FinnGen were meta-analyzed (Methods). Variants existing in at least two cohorts are shown. For variants above the top light grey dashed line ($-\log_{10}(P) > 20$), values are rescaled. The bottom dark grey dashed line indicates the genome-wide significance threshold ($P < 5.0 \times 10^{-8}$). Left panels, GRCh38-based datasets were used; variants are plotted against GRCh38 (hg38). Right panels, GRCh37-based datasets were used; variants are plotted against GRCh37 (hg19).

**Supplementary Figure 35.** Comparison of $-\log_{10}(P)$ for SNPs in GWASs based on different genome build.

a, comparison of all shared SNPs between GRCh37-based datasets and GRCh38-based datasets. The grey dashed line, a 45-degree line. Pearson correlation coefficient r is shown in the bottom right corner.
b, comparison of SNPs reaching the genome-wide significance threshold ($P < 5.0 \times 10^{-8}$).

(Methods, lines 585 – 587)

BBJ1-12K and BBJ2-42K were additionally imputed with the 1KG high-coverage reference panel⁶⁰ (GRCh38).

(Methods, lines 661 - 664)

To investigate potential associations in GRCh38-specific regions, we further conducted a cross-ancestry meta-analysis combining GRCh38-based datasets (**Supplementary Notes; Supplementary Figure 34 - 35**).

(Supplementary note)

Comparison of GWASs using GRCh38-based datasets and GRCh37-based datasets

To investigate the potential benefits of using imputation panels based on GRCh38/hg38, we processed and extracted the unrelated 2504 samples from the 1000 Genome High Coverage datasets (GRCh38/hg38) as in the 1000 Genome Phase3v5 panel (GRCh37/hg19). BBJ1-12K and BBJ2-42K were additionally imputed using the 1000 Genome panel (GRCh38/hg38). We then conducted GWAS with the same settings and additional meta-analyses using the GRCh38-based datasets and GRCh37-based datasets from BBJ1-12K, BBJ2-42K, and FinnGen for comparison. The meta-analysis using GRCh38-based datasets showed overall high consistency of $-\log_{10}(P)$ values ($r=0.92$) with GRCh37-based results and consistent genomic inflation factors λ_{gc} , with substantially high consistency for significant variants ($r=0.9991$). No additional novel loci were identified in GRCh38-specific regions. (**Supplementary Figures 34 -35**)

Line 132 - in view of the still relatively low P value for DU across ancestries, maybe "noteworthy" is a bit too strong.

We thank the reviewer for the comment and weakened the statement accordingly.

Corresponding contents in the revised manuscript:

(Results line 132-134)

The genetic correlation of GU was relatively low ($P_{gi} = 0.45$, $P = 7.3 \times 10^{-3}$), whereas the genetic architecture of DU did not show a significant difference across ancestries ($P_{gi} = 0.72$, $P = 9.6 \times 10^{-2}$; **Supplementary Table 11**).

line 145 - The authors conducted a fine-mapping analysis using SuSiE to identify the causal variants. In lower power situations, even Bayesian fine-mapping methods that simultaneously model causal variants may identify a single SNP which tags two or more causal variants and the interpretation of non-colocalisation at such false signals is likely to be misleading. Can the authors say something whether this might have influenced results?

We thank the reviewer for asking this question. We think the reviewer had described a situation where fine-mapping was performed first to identify the credible sets, and then among these signals a following pairwise colocalization analysis was employed to examine if the signals colocalize with each other, as discussed in the coloc-SuSiE paper (Wallace, C. PLoS Genet. 2021, PMID 34587156).

In this study, we conducted fine-mapping using SuSiE to identify the 95% credible sets in EAS. where we searched for nonsynonymous variants and prioritized the variants with functional annotations. Such a functionally informed method has been shown a successful approach in linking GWAS signals to missense variants (Ishigaki, K. Et al. Nat Genet. 2020, PMID 32514122; Wang, Q. S. Et al. Semin Immunopathol. 2022, PMID 35041074). Since the aim is to link PUD risk loci to potential alteration of protein functions, the following colocalization analysis was not employed in this situation. Based on the relatively large sample size for EAS-specific meta-analysis (~250,000 samples) and the use of BBJ1-180K as LD reference (N = 171,085), we think it is unlikely to be a low-power situation for this study. However, we cannot rule out the possibility that the identified variants might result from tagging distinct causal variants, which might influence the biological interpretation of such variants (partially related to comment #11 from Reviewer #1).

We revised the discussion to reflect this limitation.

(Discussion, lines 356 – 366)

Although we identified multiple associations, the current study has several potential limitations.

... Fourth, the variants identified by fine-mapping and the overlaps in association signals identified by lookup approaches might result from tagging distinct causal variants, which should be interpreted cautiously.

line 193 – “Although not statistically significant, GU showed a positive genetic correlation with GC, whereas DU was negatively correlated” – what were the (two-sided?) significance values observed?

We thank the reviewer for the comment and question. The two-sided P values for the genetic correlation are as follows: between GU and GC, 0.28; between DU and GC, 0.35. We added the source data for the plot in Figure 3a and revised the manuscript accordingly.

(Results, lines 193-195)

Although not statistically significant ($FDR < 5\%$), GU showed a positive genetic correlation with GC ($r_g=0.17$), whereas DU was negatively correlated ($r_g=-0.14$; Supplementary Table 18).

(Supplementary tables)

p1	p2	rg	se	z	p
GC	PUD	0.01106	0.1552	0.07126	9.432E-01
GC	GU	0.1702	0.1576	1.08	2.800E-01
GC	DU	-0.1486	0.1605	-0.9262	3.544E-01
GC	BU	0.01416	0.262	0.05406	9.569E-01
GC	GU_HP-	0.4454	0.4143	1.075	2.823E-01
GC	GU_HP+	0.8859	1.488	0.5955	5.515E-01
GC	HP	0.4783	0.2261	2.115	3.440E-02
GC	Age of smoking initiation	0.1625	0.197	0.8251	4.093E-01
GC	Cigarettes per day	-0.07873	0.1144	-0.688	4.915E-01
GC	Ever versus never drinkers	0.04367	0.1165	0.3747	7.079E-01
GC	Drinks per week	-0.2301	0.1078	-2.134	3.281E-02
GC	ATC_N06A	0.06157	0.227	0.2713	7.862E-01
GC	ATC_M01A	0.1253	0.2139	0.586	5.579E-01

Supplementary Table 18 (partially extracted). Genetic correlation among PUD, PUD-related phenotypes, and risk factors.

line 270 – MAGMA – there has not been a universally accepted value of the sliding window size around genes and this may need several rounds of optimization. In addition, there are some recent discussions related to the statistical stability of the H-MAGMA analysis and possible fallout (ref 35). Hence the strategy taken here requires more explanation.

We thank the reviewer for the comment on the statistical stability of MAGMA analysis in this study. To address this issue, we conducted MAGMA for gene-based, tissue-specific, and cell-specificity analyses using a range of different window sizes. As shown in **Supplementary Figure 37 - 39**, the Pearson correlation r coefficient ranges from 0.84 to 0.97 for gene-based analysis, 0.87 to 0.96 for tissue-specificity analysis, and 0.64 to 0.86 for cell-type specificity analysis, with significant associations mostly shared across different window sizes. The effect of window size selection was only marginal in this study. Based on these results, we believe MAGMA analysis showed robustness to window size selection in this study.

We added supplementary figures and revised the manuscript accordingly.

(Methods, lines 798 – 802)

We conducted pair-wise comparisons of the $-\log_{10}(p)$ values generated by MAGMA using a range of window sizes to evaluate the robustness of the association results in gene-based, tissue-type specificity, and cell-type specificity analyses discovered by MAGMA; the effect of window size selection was marginal (**Supplementary Figure 37 - 39**).

Supplementary Figure 37. Comparison of MAGMA results using different window sizes around genes for gene-based analysis.

Comparison of $-\log_{10}(P)$ values obtained by gene-based tests for PUD in EAS using different window sizes for MAGMA (implemented in FUMA). Blue marker, P value after Bonferroni correction > 0.05 for both window sizes; green marker, P value after Bonferroni correction < 0.05 for only one of the two window sizes; yellow marker, P value after Bonferroni correction < 0.05 for both window sizes. Blue line, the linear regression line with confidence interval. The sizes of squares in the upper right are proportional to the Pearson correlation coefficient r .

Supplementary Figure 38. Comparison of MAGMA results using different window sizes around genes for tissue-specificity analysis.

Comparison of $-\log_{10}(P)$ values obtained by tissue-specificity analysis for PUD in EAS using different window sizes for MAGMA (implemented in FUMA). Blue marker, FDR > 5% for both window sizes; green marker, FDR < 5% for only one of the two window sizes; yellow marker, FDR < 5% for both window sizes. Blue line, the linear regression line with confidence interval. The sizes of squares in the upper right are proportional to the Pearson correlation coefficient r .

Supplementary Figure 39. Comparison of MAGMA results using different window sizes around genes for cell-type-specificity analysis.

Comparison of meta-analyzed $-\log_{10}(P)$ values obtained by cell-type-specificity analysis for PUD using different window sizes. Blue marker, $FDR > 5\%$ for both window sizes; green marker, $FDR < 5\%$ for only one of the two window sizes; yellow marker, $FDR < 5\%$ for both window sizes. Blue line, the linear regression line with confidence interval. The sizes of squares in the upper right are proportional to the Pearson correlation coefficient r .

Discussion – In view that Hp infection is a main risk factor for PUD, how do the results relate to known risk factors for such infection (i.e. the Toll-like receptor (TLR1/6/10))? Would the difference in population be related to this? Similar points could be made for other risk factors, e.g. smoking.

We thank the reviewer for the questions on the relationship between the results of this study and previously known risk factors. Despite the well-known role of Toll-like receptors in HP infection and immunity, we did not observe risk variants or pathways directly related to TLR1/6/10 for PUD or its subtypes. We have identified signals in the MHC locus in this analysis. Due to the high complexity of the MHC region, we remain limited in our ability to further fine-map or link the signal to genes in the MHC region. (Naito, T. et al. *Semin Immunopathol.* 2021, PMID 34786601)

In this study, we conducted H.pylori (HP)-stratified analysis to investigate how the host genetic factors potentially interact with HP infection. We identified an HP-positive-specific signal for PUD at the gastrin receptor gene *CCKBR* (reported for PUD in UKB). It has been widely shown that HP-elicited cytokines stimulate gastrin release. The increased gastrin level caused by HP will likely interact with changes in *CCKBR* (rs12792379), potentially leading to dysregulated gastric acid secretion and altered susceptibility to apoptosis (Dufresne, M. et al. *Physiol Rev.* 2006. PMID 16816139; Przemec, S. M. C. et al. *Regul Pept.* 2008, PMID 17900712).

One of the major differences across ancestries identified in this study is the limited effects of risk loci at *MUC1* and *MUC6* in EAS compared with those in EUR. Despite the potential interaction of mucins with H.pylori (Navabi, N. Et al. *Infect Immun.* 2013, PMID 23275091), given the higher HP prevalence rate in East Asia than in Western Europe, this might suggest alternative mechanisms interacting with HP infection. (as discussed in Discussion lines 315 – 319)

We revised the manuscript accordingly.

(Discussion, lines 328 – 331)

It has been widely shown that HP-elicited cytokines stimulate gastrin release⁵³. It is likely that the increased gastrin level induced by HP will interact with altered expression in *CCKBR*, leading to dysregulated gastric acid secretion and altered susceptibility to apoptosis^{55,56}.

(Discussion, lines 335 - 336)

As expected, we observed high genetic correlation between GU and DU and nominally significant genetic correlations between PUD and its risk factors (Supplementary Note);

(Supplementary note)

Genetic correlation between PUD and its risk factors

It has long been known cigarette smoking increases the risk for PUD⁹. In the genetic correlation analysis, we detected a nominally significant ($P < 0.05$) negative correlation between PUD and age at smoking initiation, but such correlation was not detected for PUD and cigarettes per day, which suggest that PUD might share genetic components with long-term smoking behavior. We did not detect significant genetic correlations between PUD and drinking-related traits. The PUD risk allele for rs3859862 at *GGT1* (gamma-glutamyl transferase 1) identified in this study was linked with the alleles of a cis-eQTL and a cis-pQTL for *GGT1* that decrease the expression of *GGT1* in stomach and protein level of GGT in serum. It has been shown that alcohol consumption and smoking could lead to an increased level of GGT^{10,11}. It is worth investigating if GGT plays a role in the association between PUD and smoking/drinking. Additionally, we identified nominally significant genetic correlations of PUD/GU with chronic obstructive pulmonary disease (COPD), asthma, and rheumatoid arthritis (RA) in EAS (**Supplementary Figure 12**). A previous large-scale meta-analysis of asthma¹² identified the significant genetic correlation between asthma and PUD in addition to the significant genetic correlation of asthma with COPD or RA. The consistent observations suggested a genetic link between PUD and immune-related diseases, which has not been well studied yet and warranted further investigations.

- "Our single cell analysis further revealed the association of serotonin-secreting EC cells, somatostatin-secreting stomach D cells, and stomach tuft cells with PUD, indicating their key role in PUD etiology" - the exact (potential) roles and interactions of these cells in PUD remains very undefined in the text and could do with further elaboration.

We thank the reviewer for the comments and suggestions. We revised the discussion in the manuscript accordingly.

(Discussion, line 325)

The potential roles of D cells and EC cells were discussed in Supplementary Note.

(Supplementary note)

Potential roles of D cells and EC cells in PUD etiology

Gastrointestinal D cells are estimated to secrete ~65% circulating somatostatin, suppressing the release of gastric hormones and gastric acid^{13,14}. The PUD-associated rs2233580 in *PAX4* is a missense variant predicted to be highly deleterious (CADD > 20). It has been shown that *PAX4* is a transcriptional repressor for somatostatin¹⁵ and regulates duodenal hormone-secreting cells and serotonin/somatostatin-producing cells of the distal stomach^{16,17}. We also detected significant associations of *PDX1* in gene-based tests with independent signals (rs139276646) in its regulatory region. *PDX1* activates somatostatin transcription by interacting with its promoter¹⁸. These suggested that D cell/somatostatin dysregulation may contribute to PUD development. EC cells are the predominant source of body serotonin and play key roles in the gut-brain axis as chemosensors¹⁹, affecting a wide range of physiological processes, including gastrointestinal motility and secretion, nausea, and visceral hypersensitivity. Psychological conditions, including stress and depression, were associated with a higher risk of PUD^{20,21}. A previous large-scale study identified the causal effect of major depression (MD) on PUD²². Given the important role of serotonin in psychological conditions, the association of EC cells/serotonin with PUD, and the bidirectional effects of the brain-gut axis, it is worth further investigating whether serotonin might be a key factor in the link between depression and PUD.

Reviewer #3:

Remarks to the Author:

The authors conducted a large-scale cross-ancestry meta-analysis of genome-wide association studies of peptic ulcer disease. This was done by combining data from different population-based biobanks (Biobank Japan, Tohoku Medical Megabank, UK Biobank and FinnGen) followed by a series of downstream annotation of the identified variants based on publicly available resources. The analyses are for the most part state-of-the-art for (cross-ancestry) GWAS analyses.

We thank Reviewer #3 for the insightful comments and precise summary of this study.

Major comments:

- The majority of the samples have been included in previous published analyses which also studied peptic ulcer disease. The BBJ, FinnGen and UKBB samples have been analyzed and published as part of the large study including 220 human phenotypes published in Nature Genetics in 2021 (Sakaue et al. PMID: 34594039) and the UK Biobank and FinnGen have been queried for peptic ulcer disease specifically in a study in Nature Communications in 2021 (Wu et al. PMID: 33608531). The authors should make very clear which samples are new for the current analyses.

We thank the reviewer for the comments and apologize for the confusion.

For BBJ1-180K, this dataset was used in the discovery GWAS in this study. The BBJ1-180K dataset was also used in the large-scale study of 220 human phenotypes which reported for gastric ulcers (Sakaue et al. PMID: 34594039). Sakaue et al. did not report for other traits involved in this study based on the 7th principle in Global Biobank Meta-analysis Initiative (GBMI; <https://www.globalbiobankmeta.org/governing-principles>). Additionally, Tanikawa et al. used part of the BBJ1-180K dataset (N=32,358, whose genotyped dataset was available at that time) for GWAS of duodenal ulcer (Tanikawa, C. et al. Nature Genet. 2012, PMID 22387998). The loci identified in the previous two studies were not defined as novel in this work. Compared to prior studies using BBJ1-180K, we systematically analyzed the peptic ulcer disease (PUD; first reported for BBJ1-180K) and its subtypes, including GU, DU, and BU (first reported for BBJ1-180K) in the discovery-stage GWAS in the current study. We aim not only to analyze phenotypes not previously reported in East Asians (PUD and BU) but also to compare the genetic architectures of PUD subtypes within the same analytic framework.

For genome-wide association analyses of PUD and its subtypes in BBJ1-12K, BBJ2-42K, and TMM-50K, all the samples were used for the first time. For cross-ancestry analysis, we included only publicly available summary statistics from studies conducted in UK Biobank (Wu, Y. et al. Nat Commun. 2021, PMID 33608531; Zhou, W. et al. Nat Genet. 2018, PMID 30104761) and FinnGen (Kurki, M. I. et al. Nature. 2023, PMID 36653562). And the post-GWAS analyses mostly focused on the newly reported GWAS results from the East-Asian population.

We added a supplementary table accordingly to show the datasets used in this study.

(Supplementary tables)

Cohort	Target trait	First reported in this study	Datasets used in this study	Reference	Major differences with reference
BBJ1-180K	PUD	TRUE	Analyzed independently	/	/
BBJ1-180K	DU	FALSE	Re-analyzed independently	Tanikawa et al. PMID 22387998	a. ~127,000 additional controls b. 132 additional cases c. Imputation panel
BBJ1-180K	GU	FALSE	Re-analyzed independently	Sakaue et al. PMID 34594039	a. Imputation panel
BBJ1-180K	BU	TRUE	Analyzed independently	/	/
BBJ1-12K	PUD	TRUE	Analyzed independently	/	/
BBJ1-12K	DU	TRUE	Analyzed independently	/	/
BBJ1-12K	GU	TRUE	Analyzed independently	/	/
BBJ1-12K	BU	TRUE	Analyzed independently	/	/
BBJ1-42K	PUD	TRUE	Analyzed independently	/	/
BBJ1-42K	DU	TRUE	Analyzed independently	/	/
BBJ1-42K	GU	TRUE	Analyzed independently	/	/
BBJ1-42K	BU	TRUE	Analyzed independently	/	/
TMM-50K	PUD	TRUE	Analyzed independently	/	/
TMM-50K	DU	TRUE	Analyzed independently	/	/
TMM-50K	GU	TRUE	Analyzed independently	/	/
TMM-50K	BU	TRUE	Analyzed independently	/	/
TMM-50K	HP-stratified	TRUE	Analyzed independently	/	/
UKB	PUD	FALSE	Publicly available summary statistics	Wu et al. PMID 33608531	/
UKB	GU	FALSE	Publicly available summary statistics	Zhou et al. PMID 30104761	/
UKB	DU	FALSE	Publicly available summary statistics	Zhou et al. PMID 30104761	/
FinnGen	PUD	FALSE	Publicly available summary statistics	Kurki et al. PMID 36653562	/
FinnGen	GU	FALSE	Publicly available summary statistics	Kurki et al. PMID 36653562	/
FinnGen	DU	FALSE	Publicly available summary statistics	Kurki et al. PMID 36653562	/

Supplementary Table 32. Summary of cohorts and phenotypes analyzed in the genome-wide association analysis in this study.

(Methods, lines 553 – 555)

The cohorts and phenotypes that were first reported in this study were summarized in **Supplementary Table 32**.

- The case definition in the TMM-50K is based on self-reported PUD. While individuals might potentially reliably report peptic ulcer disease, the distinction between gastric or duodenal ulcers will be very unreliable if based on questionnaires.

We thank the reviewer for the comment and concern. In this study, the phenotypes (DU and GU) were based on clinical interviews or questionnaires instead of clinical tests or endoscopy, which are the gold standard for diagnosis but are usually impractical to conduct for screening in biobank-scale cohorts.

We totally agree with the reviewer that finer phenotyping by clinical tests or endoscopy could be more reliable, which could subsequently be one of the key factors that lead to the elucidation of

the heterogeneity of PUD, especially for GU, as we identified in this work. As indicated in the discussion section, we noted that this is one of the limitations of this study.

Despite the self-reported phenotypes in BBJ and TMM, we successfully replicated most of the previously known loci identified in European individuals (7 out of 8 loci), and the novel biological findings in this study are feasible. These suggested relatively high reliability of the results.

To reflect this, we rephrased the texts in the manuscript:

(Discussion, lines 356 - 370)

Although we identified multiple associations, the current study has several potential limitations. First, the phenotypic information of PUD and subtypes was obtained via interviews and reviews of medical records. However, the prevalence rate of PUD was consistent with that in previous epidemiological studies; **our study replicated most of the previously identified loci, and the novel biological findings are feasible, which suggested the relatively high reliability of the results.** ... Even though our subtype analysis revealed the overall similarities and differences in genetic architecture, a large sample size and more detailed classifications are still warranted to elucidate the potential heterogeneity further.

- The major causes for peptic ulcer disease are Helicobacter pylori infection and the use of NSAIDs. The authors do perform a H. pylori stratified analyses, but this is only done in the TMM-50K cohort (based on self-reported PUD) and not in the other cohorts. The diagnosis of PUD in the BBJ cohorts is made based on interviews and reviewing medical records. Is there more detail on the cause of peptic ulcer disease available, e.g., HP or NSAID?

We thank the reviewer for the comment and the important question. In this work, we only conducted the H.pylori (HP)-stratified analysis in the TMM-50K cohort based on serum anti-HP IgG levels. Since serum anti-HP IgG level was unavailable in BBJ, we did not conduct HP-stratified analysis in BBJ cohorts in this study. We would like to note that as far as we are concerned, the HP-stratified analysis in this study was the first large-scale HP-stratified GWAS by far. Even though HP-stratified analysis was only conducted in TMM-50K, the findings were likely due to biological mechanisms instead of lack of power, based on the power analysis (related to Reviewer #1's comment).

Although BBJ and TMM collected drug prescription records which include prescriptions of NSAIDs, we remain limited in the ability to determine the effect of NSAIDs on PUD. Due to our limited access to electronic health records (EHR) and lack of information regarding the chronological order of PUD onset and NSAID use at PUD onset in this study, the specific interaction of NSAIDs use with host genetic factors was not investigated in this study.

The limitations of this study were reflected in the discussion:

(Discussion, lines 356 - 370)

Although we identified multiple associations, the current study has several potential limitations. First, the phenotypic information of PUD and subtypes was obtained via interviews and reviews of medical records. However, the prevalence rate of PUD was consistent with that in previous epidemiological studies; **our study replicated most of the previously identified loci, and the novel biological findings are feasible, which suggested the relatively high reliability of the results.** Second, due to the lack of information regarding the chronological order of PUD onset and anti-inflammatory drug use at PUD onset in this study, the specific interaction of NSAIDs with host genetic factors was under-explored. Third, detailed information on the anatomic site of the ulcers or the strains of HP was not available. ... Even though our subtype analysis revealed the overall similarities and differences in genetic architecture, a large sample size and more detailed classifications are still warranted to elucidate the potential heterogeneity further.

- Overall, they identify 25 novel loci that are concordant across ancestries. But the manuscript is by times difficult to read, listing numbers of associated genetic variants for many different analyses, e.g. in PUD, GU, DU, and BU either in East Asian populations or in combined populations. It would be helpful to present these results better. Maybe in a figure that makes clear which result come from which analyses and which results are novel.

We thank the reviewer for the suggestion. We agree with the reviewer and accordingly created an annotated ideogram using PhenoGram (Wolfe, D. et al. BioData Min. 2013, PMID 24131735; **Supplementary Figure 6**) to present the novel loci. This figure specified the analysis that identified the significant locus.

We added a supplementary figure and modified texts in the manuscript accordingly.

(Supplementary figures)

Supplementary Figure 6. Phenogram of genome-wide significant loci for PUD and its subtypes.

The shapes indicate ancestry: circles, cross-ancestry (Cross-ancestry); diamonds, East Asians (EAS); triangles, Europeans (EUR). Colors indicate peptic ulcer disease (PUD) and its subtypes: blue, PUD; green, Duodenal ulcers (DU); red, Gastric Ulcers (GU); black, Both gastric and duodenal ulcers (BU). Identified loci are annotated with the nearest genes to the most significant lead variants. Novel loci were indicated by bold and underlined gene names. Cytobands and annotations are based on GRCh37/hg19. Chromosomes with no associated loci are omitted.

(Results, lines 112 - 114)

In total, we identified 25 non-overlapping novel loci for PUD and its subtypes in the East Asian-specific and cross-ancestry meta-analyses (Supplementary Figure 6), ...

- The paper is heavily focused on the computational analyses and downstream annotation. It lacks a good discussion on the biological interpretation of the identified variants and potential implications.

We thank the reviewer for pointing out this. We revised the discussion section to reflect the key biological interpretations.

(Discussion, lines 325 - 333)

The potential roles of D cells and EC cells were discussed in Supplementary Note. Our results also showed the signal at *CCKBR* (receptor for gastrin) to be HP-positive-specific. The PUD risk allele of the lead SNP (rs12792379) is in LD with the eQTL allele associated with higher *CCKBR* expression in multiple tissues²¹, including esophagus mucosa. It has been widely shown that HP-elicited cytokines stimulate gastrin release⁵³. It is likely that the increased gastrin level induced by HP will interact with altered expression in *CCKBR*, leading to dysregulated gastric acid secretion and altered susceptibility to apoptosis^{55,56}. Taken together, our results provided genetic evidence of gastrointestinal cell differentiation and hormone regulation being critical in PUD etiology.

(Supplementary note)

Potential roles of D cells and EC cells in PUD etiology

Gastrointestinal D cells are estimated to secrete ~65% circulating somatostatin, suppressing the release of gastric hormones and gastric acid^{13,14}. The PUD-associated rs2233580 in *PAX4* is a missense variant predicted to be highly deleterious (CADD > 20). It has been shown that *PAX4* is a transcriptional repressor for somatostatin¹⁵ and regulates duodenal hormone-secreting cells and serotonin/somatostatin-producing cells of the distal stomach^{16,17}. We also detected significant associations of *PDX1* in gene-based tests with independent signals (rs139276646) in its regulatory region. *PDX1* activates somatostatin transcription by interacting with its promoter¹⁸. These suggested that D cell/somatostatin dysregulation may contribute to PUD development. EC cells are the predominant source of body serotonin and play key roles in the gut-brain axis as chemosensors¹⁹, affecting a wide range of physiological processes, including gastrointestinal motility and secretion,

nausea, and visceral hypersensitivity. Psychological conditions, including stress and depression, were associated with a higher risk of PUD^{20,21}. A previous large-scale study identified the causal effect of major depression (MD) on PUD²². Given the important role of serotonin in psychological conditions, the association of EC cells/serotonin with PUD, and the bidirectional effects of the brain-gut axis, it is worth further investigating whether serotonin might be a key factor in the link between depression and PUD.

Decision Letter, first revision:

Our ref: NG-A61277R

2nd May 2023

Dear Yoichiro,

Your revised manuscript "East Asian-specific and cross-ancestry genome-wide meta-analyses provide mechanistic insights into peptic ulcer disease" (NG-A61277R) has been seen by the original referees. As you will see from their comments below, they find that the paper has improved in revision, and therefore we will be happy in principle to publish it in Nature Genetics as an Article pending final revisions to satisfy the referees' remaining requests and to comply with our editorial and formatting guidelines.

We are now performing detailed checks on your paper, and we will send you a checklist detailing our editorial and formatting requirements soon. Please do not upload the final materials or make any revisions until you receive this additional information from us.

Thank you again for your interest in Nature Genetics. Please do not hesitate to contact me if you have any questions.

Sincerely,
Kyle

Kyle Vogan, PhD
Senior Editor
Nature Genetics
<https://orcid.org/0000-0001-9565-9665>

Reviewer #1 (Remarks to the Author):

The authors have done a great job responding to my comments. I'm satisfied with the current manuscript, with the following two minor comments:

1. In the fine-mapping analysis, the authors used the summary statistics of meta-analysis, including BBJ1-180K, BBJ1-12K, BBJ2-42K, and TMM-50K, and the LD from unrelated individuals from only

BBJ1-180K. Kanai et al. Cell Genomics 2022

(<https://www.sciencedirect.com/science/article/pii/S2666979X2200163X>) warned about a possible miscalibration if LD is not calculated from the exact study sample. Given that BBJ1-180K has over half of the samples in the study and the expected similarity across cohorts, I don't think this issue will be severe, but it'd be good if the authors consider adding this as a limitation.

2. I wonder if the authors could add numbers to support this claim: "Our results also showed the signal at CCKBR (receptor for gastrin) to be HP-positive-specific." For example, what's the effect size with confidence interval for the signal at CCKBR for HP-positive and HP-negative respectively?

Reviewer #2 (Remarks to the Author):

The authors conducted a GWAS of peptic ulcer disease in Asian ancestry and compare results to European ancestries and others. Many novel loci associated with disease are identified. In general, this is an important study, enhancing our understanding of the disease and I support its publication in the present form.

Reviewer #3 (Remarks to the Author):

The comments have been addressed in a satisfactory manner by the authors.

I have one remaining comment:

The authors identify 25 novel loci that are concordant across ancestries. But it remains difficult to keep track of which variant is identified in which analysis. (The authors list numbers of associated genetic variants for many different analyses, e.g. in PUD, GU, DU, and BU either in East Asian populations or in combined populations.)

To address this issue in the resubmission, the authors created a figure which is now in the supplement as Supplementary Figure 6. "Phenogram of genome-wide significant loci for PUD and its subtypes. "

For clarity and interpretation of the results by the readers, I would consider to move this overview figure into the main manuscript.

Author Rebuttal, first revision:

Response

Reviewer #1:

Remarks to the Author:

The authors have done a great job responding to my comments. I'm satisfied with the current manuscript, with the following two minor comments:

1. In the fine-mapping analysis, the authors used the summary statistics of meta-analysis, including BBJ1-180K, BBJ1-12K, BBJ2-42K, and TMM-50K, and the LD from unrelated individuals from only BBJ1-180K. Kanai et al. Cell Genomics 2022 (<https://www.sciencedirect.com/science/article/pii/S2666979X2200163X>) warned about a possible miscalibration if LD is not calculated from the exact study sample. Given that BBJ1-180K has over half of the samples in the study and the expected similarity across cohorts, I don't think this issue will be severe, but it'd be good if the authors consider adding this as a limitation.

We thank the reviewer for pointing this out. We agree with the reviewer on the potential miscalibration resulting from not using the exact same samples for LD calculation. To reflect this point, we revised the manuscript to add this as a potential limitation.

Revised manuscript:

(Discussion, lines 381 - 384)

Fourth, the variants identified by fine-mapping and the overlaps in association signals identified by lookup approaches might result from tagging distinct causal variants, and the meta-analysis fine-mapping using BBJ1-180K as LD reference might be miscalibrated⁶², which should be interpreted cautiously.

2. I wonder if the authors could add numbers to support this claim: "Our results also showed the signal at CCKBR (receptor for gastrin) to be HP-positive-specific." For example, what's the effect size with confidence interval for the signal at CCKBR for HP-positive and HP-negative respectively?

We thank the reviewer for the suggestion. We added the statistics for the effect sizes in the manuscript.

Revised manuscript:

(Results, lines 225 - 229)

We identified one lead SNP (rs12792379), specifically associated with HP-positive PUD, at *CCKBR* (OR= 1.18, CI = 1.05 – 1.34 for HP-positive PUD; OR = 1.01, CI= 0.92 - 1.11 for HP-negative PUD), ...

Reviewer #2:

Remarks to the Author:

The authors conducted a GWAS of peptic ulcer disease in Asian ancestry and compare results to European ancestries and others. Many novel loci associated with disease are identified. In general, this is an important study, enhancing our understanding of the disease and I support its publication in the present form.

We thank the reviewer for the constructive comments and the kind support for the publication of this work.

Reviewer #3:

Remarks to the Author:

The comments have been addressed in a satisfactory manner by the authors

I have one remaining comment:

The authors identify 25 novel loci that are concordant across ancestries. But it remains difficult to keep track of which variant is identified in which analysis. (the authors list , numbers of associated genetic variants for many different analyses, e.g. in PUD, GU, DU, and BU either in East Asian populations or in combined populations.)

To address this issue in the resubmission the authors created a figure which is now in the supplement as Supplementary Figure 6. "Phenogram of genome-wide significant loci for PUD and its subtypes. "

For clarity and interpretation of the results by the readers I would consider to move this overview figure into the main manuscript.

We thank the reviewer for the suggestion on the clarity and interpretation of the results. To improve this, we previously created a phenogram as Supplementary Figure 6. We have carefully considered your further suggestions and those of the editor, based on which we agree that Supplementary Figure 6 is best suited as a supplementary figure since the information in Supplementary Figure 6 is mostly redundant with the data presented in Table 1 in the main text and Supplementary Table 2, 6 and 8. We understand your suggestion to elevate the importance of Supplementary Figure 6, but we respectfully believe that its current placement as a supplementary figure adequately serves its purpose and also ensures that the main text remains concise and less redundant. Again, we sincerely appreciate your constructive feedback and suggestions.

Final Decision Letter:

12th Oct 2023

Dear Dr. Kamatani,

I am delighted to say that your manuscript "East-Asian-specific and cross-ancestry genome-wide meta-analyses provide mechanistic insights into peptic ulcer disease" has been accepted for publication in an upcoming issue of Nature Genetics.

Your paper will be published online after we receive your corrections and will appear in print in the next available issue. You can find out your date of online publication by contacting the Nature Press Office (press@nature.com) after sending your e-proof corrections. Now is the time to inform your Public Relations or Press Office about your paper, as they might be interested in promoting its publication. This will allow them time to prepare an accurate and satisfactory press release. Include your manuscript tracking number (NG-A61277R1) and the name of the journal, which they will need when they contact our Press Office.

Please note that *Nature Genetics* is a Transformative Journal (TJ). Authors may publish their research with us through the traditional subscription access route or make their paper immediately open access through payment of an article-processing charge (APC). Authors will not be required to make a final decision about access to their article until it has been accepted. [Find out more about Transformative Journals](https://www.springernature.com/gp/open-research/transformative-journals)

Authors may need to take specific actions to achieve [compliance with funder and institutional open access mandates](https://www.springernature.com/gp/open-research/funding/policy-compliance-faqs). If your research is supported by a funder that requires immediate open access (e.g. according to [Plan S principles](https://www.springernature.com/gp/open-research/plan-s-compliance)) then you should select the gold OA route, and we will direct you to the compliant route where possible. For authors selecting the subscription publication route, the journal's standard licensing terms will need to be accepted, including [self-archiving-and-license-to-publish](https://www.nature.com/nature-portfolio/editorial-policies/self-archiving-and-license-to-publish). Those licensing terms will supersede any other terms that the author or any third party may assert apply to any version of the manuscript.

To assist our authors in disseminating their research to the broader community, our SharedIt initiative provides you with a unique shareable link that will allow anyone (with or without a subscription) to read the published article. Recipients of the link with a subscription will also be able to download and

print the PDF.

If you have not already done so, we invite you to upload the step-by-step protocols used in this manuscript to the Protocols Exchange, part of our on-line web resource, natureprotocols.com. If you complete the upload by the time you receive your manuscript proofs, we can insert links in your article that lead directly to the protocol details. Your protocol will be made freely available upon publication of your paper. By participating in natureprotocols.com, you are enabling researchers to more readily reproduce or adapt the methodology you use. [Natureprotocols.com](http://natureprotocols.com) is fully searchable, providing your protocols and paper with increased utility and visibility. Please submit your protocol to <https://protocolexchange.researchsquare.com/>. After entering your nature.com username and password you will need to enter your manuscript number (NG-A61277R1). Further information can be found at <https://www.nature.com/nature-portfolio/editorial-policies/reporting-standards#protocols>

Sincerely,

Wei

Wei Li, PhD
Senior Editor
Nature Genetics
New York, NY 10004, USA
www.nature.com/ng